
# Gas-phase chemistry in the online multiscale NMMB/BSC Chemical Transport Model: Description and evaluation at global scale

Alba Badia[1,*], Oriol Jorba[1], Apostolos Voulgarakis[2], Donald Dabdub[3], Carlos Pérez García-Pando[4, 5], Andreas Hilboll[6,7], María Gonçalves[1, 8], and Zavisa Janjic[9]

[1]Earth Sciences Department, Barcelona Supercomputing Center-Centro Nacional de Supercomputación, Barcelona, Spain
[2]Department of Physics, Imperial College, London
[3]Mechanical and Aerospace Engineering, University of California, Irvine
[4]NASA Goddard Institute for Space Studies, New York, USA
[5]Department of Applied Physics and Applied Math, Columbia University, New York, USA
[6]Institute of Environmental Physics, University of Bremen, Germany
[7]MARUM – Center for Marine Environmental Sciences, University of Bremen, Germany
[8]Universitat Politecnica de Catalunya, Barcelona, Spain
[9]National Centers for Environmental Prediction, College Park, MD, USA
[*]Now in the Centre for Ocean and Atmospheric Sciences, School of Environmental Sciences, University of East Anglia, Norwich, United Kingdom

*Correspondence to:* oriol.jorba@bsc.es

**Abstract.**

This paper presents a comprehensive description and benchmark evaluation of the tropospheric gas-phase chemistry component of the NMMB/BSC Chemical Transport Model (NMMB/BSC-CTM), an online chemical weather prediction system conceived for both the regional and the global scale. We provide an extensive evaluation of a global annual cycle simulation us-

5  ing a variety of background surface stations (EMEP, WDCGG and CASTNET), ozonesondes (WOUDC, CMD and SHADOZ), aircraft data (MOZAIC and several campaigns), and satellite observations (SCIAMACHY and MOPITT). We also include an extensive discussion of our results in comparison to other state-of-the-art models.

The model shows a realistic oxidative capacity across the globe. The seasonal cycle for CO is fairly well represented at different locations (correlations around 0.3-0.7 in surface concentrations), although concentrations are underestimated in spring and

10  winter in the Northern Hemisphere, and are overestimated throughout the year at 800 and 500 hPa in the Southern Hemisphere.

Nitrogen species are well represented in almost all locations, particularly $NO_2$ in Europe (RMSE below 9 $\mu$g $m^{-3}$). The modeled vertical distribution of $NO_x$ and $HNO_3$ are in excellent agreement with the observed values and the spatial and seasonal trends of tropospheric $NO_2$ columns correspond well to observations from SCIAMACHY, capturing the highly polluted areas and the biomass burning cycle throughout the year. Over Asia, the model underestimates $NO_x$ from March to August

15  probably due to an underestimation of $NO_x$ emissions in the region. Overall, the comparison of the modelled CO and $NO_2$ with MOPITT and SCIAMACHY observations emphasizes the need for more accurate emission rates from anthropogenic and biomass burning sources (i.e., specification of temporal variability).

The resulting ozone ($O_3$) burden (348 Tg) lies within the range of other state-of-the-art global atmospheric chemistry models. The model generally captures the spatial and seasonal trends of background surface $O_3$ and its vertical distribution. However,





the model tends to overestimate $O_3$ throughout the troposphere in several stations. This is attributed to an overestimation of CO concentration over the southern hemisphere leading to an excessive production of $O_3$. Overall, $O_3$ correlations range between 0.6 to 0.8 for daily mean values. The overall performance of the NMMB/BSC-CTM is comparable to that of other state-of-the-art global chemical transport models.

## 1  Introduction

Tropospheric ozone ($O_3$) is a radiatively active gas interacting with solar and terrestrial radiation that is mainly produced during the photochemical oxidation of methane ($CH_4$), carbon monoxide (CO) and non-methane volatile organic compounds (NMVOC) in the presence of nitrogen oxides ($NO_x$) (Crutzen, 1974; Derwent et al., 1996). Downward transport from the stratosphere, where $O_3$ is created by photolysis of oxygen ($O_2$) molecules, is also an important source of tropospheric $O_3$

(Stohl et al., 2003; Hsu and Prather, 2009). In urban areas, $O_3$ is a major component of 'smog', which can cause a number of respiratory health effects (WHO, 2014). Since the pre-industrial era, changes in emissions of $O_3$ precursors from anthropogenic and biomass burning sources have modified the distribution of tropospheric $O_3$ and other trace gases (Lamarque et al., 2013). Tropospheric $O_3$, with an average lifetime of the order of weeks, is highly variable in space and time, and Air Quality Models (AQM) are required to predict harmful levels of $O_3$ along with its precursors and other trace gases.

AQMs are driven by meteorological fields and fed by emission inventories. They include a chemical mechanism for representing gas-phase and aerosol atmospheric chemistry, a photolysis scheme describing the photo-dissociation reactions driven by sunlight, dry and wet deposition schemes to account for the removal of pollutants from the atmosphere, and the characterization of the downward transport of stratospheric $O_3$. The development of AQMs and meteorological models (MM) evolved as separate fields (offline approach) due to complexity and limitations in computer resources.

The offline approach requires lower computational capacity, but involves a loss of essential information on atmospheric processes whose time-scale is smaller than the output time rate of the meteorological model (Baklanov et al., 2014). Nowadays, owing to a general increase in computer capacity, online coupled meteorology-chemistry models are being increasingly developed and used by the scientific community, who recognizes the advantages of the online approach (Byun, 1990). Overviews of online AQM-MM models are available in the literature (Zhang, 2008; Baklanov et al., 2014).

Several global AQMs have been developed during the last decades: online multiscale GEM-AQ (1.5° x 1.5°) (Gong et al., 2012), offline TM5-chem-v3.0 (3° x 2°) (Huijnen et al., 2010), online LMDZ-INCA (3.8° x 2.5°) (Folberth et al., 2006), online GATOR-GCMM (4° x 5°) (Jacobson, 2001), online IFS-MOZART used in MACC project (80km x 80km) (Inness et al., 2013), online C-IFS recently developed at ECMWF (80km x 80km) (Flemming et al., 2015), and offline MOZART-4 (2.8° x 2.8°) (Emmons et al., 2010). Most of these models have been applied at coarse resolutions with simplified chemical

schemes. Currently, the systems are being updated and prepared for higher resolution applications.

In this contribution, we describe and evaluate the gas-phase chemistry of the NMMB/BSC Chemical Transport Model (NMMB/BSC-CTM), an online multi-scale non-hydrostatic chemical weather prediction system that can be run either globally





or regionally (Pérez et al., 2011; Jorba et al., 2012). We provide a thorough evaluation over a one-year period for the global domain using an horizontal resolution of $1°$ x $1.4°$.

The NMMB/BSC-CTM model, configured as a limited area model, has recently participated in the Air Quality Model Evaluation International Initiative Phase2 (AQMEII-Phase2) intercomparison exercise. A spatial, temporal and vertical evaluation

of the chemical model results for the year 2010 on a regional scale are presented in Badia and Jorba (2014). Moreover, a comparison between other modeling systems currently applied in Europe and North America in the context of AQMEII phase 2 is presented in Im et al. (2014). Evaluations of previous version of the model include the dust implementation, presented in Pérez et al. (2011) and Haustein et al. (2012), and the sea-salt aerosol module, described and evaluated at the global scale in Spada et al. (2013) and Spada et al. (2015). The aerosol module for other relevant global aerosols (natural, anthropogenic and sec-

ondary) is currently under development within the NMMB/BSC-CTM (Spada et al., in prep). This initiative aims at developing a fully coupled chemical multiscale (global/regional) weather prediction system that resolves gas-aerosol-meteorology interactions and provides initial and boundary conditions for embedded high resolution nests in a unified dynamics-physics-chemistry environment.

The focus of this paper is to describe and evaluate the global atmospheric model NMMB/BSC-CTM in terms of the spatial

distribution and seasonal variations of $O_3$ and its precursors. In Sec. 2, we provide a description of the atmospheric driver, the gas-phase chemistry module, and the model configuration including the online biogenic emissions. Section 3 presents an overview of the model setup with a description of the chemical and meteorological initial conditions, and the anthropogenic and biomass burning emissions implemented for this experiment. We illustrate the capability of the NMMB/BSC-CTM to reproduce the main reactions occurring in the atmosphere by evaluating the model with ground-based monitoring stations,

ozonesondes, aircraft data, climatological vertical profiles and satellite retrievals, which are described in Sec. 4. The results of the model performance are discussed in Sec. 5 for year 2004. The last section summarizes the conclusions of this work.

## 2   Model description

The NMMB/BSC-CTM is a fully online multiscale chemical transport model for regional and global-scale applications (Pérez et al., 2011; Jorba et al., 2012). The system is based on the meteorological Non-hydrostatic Multiscale Model on B-grid

(NMMB; Janjic and Gall, 2012), developed at the National Centers for Environmental Prediction (NCEP). The model couples online the NMMB with the gas-phase and aerosol continuity equations to solve the atmospheric chemistry processes with detail. Due to its online-coupling approach, the model accounts for the feedback processes of gases, aerosol particles and radiation. Currently, it can consider the radiative effect of aerosols while presently ignoring cloud-aerosol interactions. In the present work, only the gas-phase chemistry is used, thus no interaction between gas-phase and aerosol-phase is applied. In this

section we provide a concise description of the NMMB model and the gas-phase chemistry module of the BSC-CTM.



## 2.1 The Non-hydrostatic Multiscale Model on B-grid

The Non-hydrostatic Multiscale Model on B grid(NMMB; Janjic and Black, 2005; Janjic and Gall, 2012) was conceived for short- and medium-range forecasting over a wide range of spatial and temporal scales from large eddy simulations (LES) to global simulations. Its unified non-hydrostatic dynamical core allows regional and global simulations, including embedded regional nests. The NMMB has been developed within the Earth System Modeling Framework (ESMF) at NCEP, following the general modeling philosophy of the NCEP regional Weather Research and Forecasting (WRF) Non-hydrostatic Mesoscale Model (NMM; Janjic et al., 2001; Janjic, 2003). The regional NMMB has been the operational regional North American Mesoscale (NAM) model at NCEP since October 2011. The numerical schemes used in the model were designed following the principles presented in Janjic (1977, 1979, 1984, 2003). Isotropic horizontal finite volume differencing is employed so a variety of basic and derived dynamical and quadratic quantities are conserved. Among these, the conservation of energy and entrophy (Arakawa, 1966) improves the accuracy of the nonlinear dynamics. The hybrid pressure-sigma coordinate is used in the vertical direction and the Arakawa B-grid is applied in the horizontal direction. The global model on the latitude-longitude grid with polar filtering was developed as the reference version, and other geometries such the cubed-sphere are currently being tested. The regional model is formulated on a rotated longitude-latitude grid, with the Equator of the rotated system running through the middle of the integration domain resulting in more uniform grid distances. The non-hydrostatic component of the model dynamics is introduced through an add-on module that can be turned on or off, depending on the resolution. The operational physical package includes: (1) the Mellor-Yamada-Janjic (MYJ) level 2.5 turbulence closure for the treatment of turbulence in the planetary boundary layer (PBL) and in the free atmosphere (Janjic et al., 2001), (2) the surface layer scheme based on the Monin-Obukhov similarity theory (Monin and Obukhov, 1954) with introduced viscous sublayer over land and water (Zilitinkevich, 1965; Janjic, 1994), (3) the NCEP NOAH (Ek et al., 2003) or the LISS land surface model (Vukovic and Z., 2010) for the computation of the heat and moisture surface fluxes, (4) the GFDL or RRTMG long-wave and shortwave radiation package (Fels and Schwarzkopf, 1975; Mlawer et al., 1997), (5) the Ferrier gridscale clouds and microphysics (Ferrier et al., 2002), and (6) the Betts-Miller-Janjic convective parametrization (Betts, 1986; Betts and Miller, 1986; Janjic, 1994, 2000). Vertical diffusion is handled by the surface layer scheme and by the PBL scheme. Lateral diffusion is formulated following the Smagorinsky non-linear approach (Janjic, 1990).

## 2.2 Gas-phase chemistry module

The tropospheric gas-phase chemistry module is coupled online within the NMMB code. Different chemical processes were implemented following a modular operator splitting approach to solve the advection, diffusion, chemistry, dry and wet deposition, and emission processes. Meteorological information is available at each time step to solve the chemical processes properly. In order to maintain consistency with the meteorological solver, the chemical species are advected and mixed at the corresponding time step of the meteorological tracers using the same numerical schemes implemented in the NMMB. The advection scheme is Eulerian, positive definite and monotone, maintaining a consistent mass-conservation of the chemical species within the domain of study (Janjic et al., 2009; Tang et al., 2009; Janjic and Gall, 2012).





### 2.2.1 Chemical-phase reaction mechanism

Several chemical mechanisms can be implemented within the NMMB/BSC-CTM. A modular coupling with the Kinetic Pre-Processor (KPP) package (Damian et al., 2002; Sandu and Sander, 2006) allows the model to maintain wide flexibility. Additionally, an Eulerian-Backward-Iterative solver (Hertel et al., 1993) was implemented as a complementary option to the KPP

solvers to allow the model to run with a fast ordinary differential equation solver at global scales. For the present study, we use a Carbon-Bond family mechanism, the Carbon Bond 2005 (CB05; Yarwood et al., 2005), an updated version of the Carbon-Bond IV (CB4) lumped-structure-type mechanism (Gery et al., 1989). CB4 was formulated focusing on limited domain extent, urban and regional environments and for planetary boundary layer chemistry. CB05 extends its applicability from urban to remote tropospheric conditions and is suitable for global applications. CB05 was evaluated against smog chamber data from the

University of California, Riverside and University of North Carolina. It includes 51 chemical species and solves 156 reactions (see Tables S1 and S2 in the supplementary information). Both the organic chemistry of methane and ethane, and the chemistry of methylperoxy radical, methyl hydroperoxide and formic acid are treated explicitly. The higher organic peroxides, organic acids, and peracids are treated as lumped species. Following its main design, CB05 defines proxy single and double carbon bond species, paraffin and an olefin bond respectively, and it introduces the internal olefin species. The rate constants were

updated based on evaluations from Atkinson et al. (2004) and Sander et al. (2006). Organic compounds not explicitly treated are apportioned to the carbon-bond species based on the molecular structure and following Yarwood et al. (2005) assignments from VOC species to CB05 model species. The concentration of methane is considered constant (1.85 ppm) in this study.

### 2.2.2 Photolysis scheme

One of the most important processes determining tropospheric composition is the photo-dissociation of trace gases. Table S2

in the supplementary information lists the photolysis reactions considered. To compute the photolysis rates, we implemented the Fast-J (Wild et al., 2000) online photolysis scheme. Fast-J has been coupled with the physics of each model layer (e.g., clouds and absorbers such as $O_3$). The optical depths of grid-scale clouds from the atmospheric driver are considered by using the fractional cloudiness based on relative humidity (Fast et al., 2006). The main advantages of Fast-J are the optimization of the phase function expansion into Legendre polynomials and the optimization of the integration over wavelength (Wild et al.,

2000). The Fast-J scheme has been upgraded with CB05 photolytic reactions. The quantum yields and cross section for the CB05 photolysis reactions have been revised and updated following the recommendations of Atkinson et al. (2004) and Sander et al. (2006). The Fast-J scheme uses seven different wavelength bins appropriate for the troposphere to calculate the actinic flux covering from 289 to 850 nm (see Table VIII from Wild et al. (2000)). In this work, aerosols are not considered in the photolysis rate calculation.

### 2.2.3 Dry-deposition scheme

The dry-deposition scheme is responsible for computing the flux of trace gases from the atmosphere to the surface. It is calculated by multiplying concentrations in the lowest model layer by the spatially and temporally varying deposition velocity:


$$\frac{\partial C_i}{\partial t}_{dry-dep} = -C_i v_d \tag{1}$$

where $t$ is the time, $i$ the gas-phase species, $C_i$ is the concentration of the gas in the lowest model layer, and $v_d$ is the dry-deposition velocity. At each time step, $v_d$ is calculated according to:

$$|v_d| = \frac{1}{(R_a + R_b + R_c)} \tag{2}$$

where $R_a$ is the aerodynamic resistance (depends only on atmospheric conditions), $R_b$ is the quasilaminar sublayer resistance (depends on friction velocity and molecular characteristics of gases), and $R_c$ is the canopy or surface resistance (depends on surface properties and the reactivity of the gas). $R_a$ and $R_b$ are computed following their common definition (Seinfeld and Pandis, 1998), while $R_c$ is simulated following Wesely (1989), where the surface resistance is derived from the resistances of the surfaces of the soil and the plants. The properties of the plants are determined using land-use data (from the meteorological driver USGS land-use) and depend on the season. The surface resistance also depends on the diffusion coefficient, the reactivity, and water solubility of the reactive trace gases.

### 2.2.4 Wet-deposition scheme

We use the scheme of Byun and Ching (1999) and Foley et al. (2010) to resolve the cloud processes affecting the concentration of 36 gases from the CB05 chemical mechanism. The processes included are grid-scale scavenging and wet-deposition, subgrid-scale vertical mixing, and scavenging and wet-deposition for precipitating and non-precipitating clouds. At the moment, we consider only in-cloud scavenging, which is computed using the Henry's Law equilibrium equation. The rate of change for in-cloud pollutant concentration is given by:

$$\frac{\partial C_{icld}}{\partial t} = C_{icld}\frac{e^{-\alpha_i \tau_{cld}} - 1}{\tau_{cld}} \tag{3}$$

where $C_{icld}$ is the gas concentration within the cloud [ppm], $\tau_{cld}$ is the cloud timescale [s], and $\alpha_i$ is the scavenging coefficient for the gas species that is calculated as:

$$\alpha_i = \frac{1}{\tau_{washout}(1 + \frac{TWF}{H_i})}, \tag{4}$$

where H$_i$ is the Henry's Law coefficient for the gas species [M/atm], TWF=$\rho_{H_2O}/(W_T RT)$ is the total water fraction (where $W_T$ is the total mean water content [M/atm], $R$ is the Universal gas constant, and $T$ is the in-cloud air temperature [K]), and $\tau_{washout}$ is the washout time [s], i.e., the amount of time required to remove all of the water from the cloud volume at a specified precipitation rate [m/s], which is given by:

$$\tau_{washout} = \frac{W_T \Delta Z_{cld}}{\rho_{H_2O} P_r} \tag{5}$$

where $\Delta Z_{cld}$ is the cloud thickness [m] and $P_r$ is the precipitation rate [m/s]. Both grid-scale and subgrid-scale scavenging are computed with equation 3, where $\tau_{cld}$ is 1 hour for subgrid-scale clouds, and the chemistry timestep for grid-scale clouds. Wet deposition is computed following the algorithm of Chang et al. (1987), which depends upon $P_r$ and the gas concentration within the cloud $C_{icld}$. Thus, the wet deposition is given by:

$$wdep_i = \int\limits_0^{\tau_{cld}} C_{icld} P_r dt \tag{6}$$

The sub-grid cloud scheme implemented solves convective mixing, scavenging and wet deposition of a representative cloud within the grid cell following the CMAQ and RADMv2.6 model schemes (Byun and Ching, 1999; Chang et al., 1987). Precipitating and non-precipitating sub-grid clouds are considered. The latter are categorized as pure fair weather clouds and non-precipitating clouds and may coexist with precipitating clouds (Byun and Ching, 1999; Foley et al., 2010).

### 2.2.5 Upper boundary conditions

Because the model focuses on the troposphere, stratospheric chemistry is taken into account using a simplified approach. Above 100 hPa, mixing ratios of several species (NO, $NO_2$, $N_2O_5$, $HNO_3$ and CO) are initialized each day from a global chemical model MOZART-4 (Emmons et al., 2010). For $O_3$, an important reactive gas requiring a more refined representation in the stratosphere, we use a linear $O_3$ stratospheric scheme, COPCAT (Monge-Sanz et al., 2011). COPCAT is based on the approach of Cariolle and Déqué (1986), which represented the first effort to include a linearized $O_3$ scheme (named Cariolle v1.0) in a three-dimensional model.

Following the COPCAT approach, the change in $O_3$ with time due to local chemistry is given by:

$$\frac{\partial C_{O_3}}{\partial t} = \frac{d\chi}{dt} = (P - L)[\chi, T, \Phi] \tag{7}$$

where $(P - L)$ represents the $O_3$ tendency as a linear function depending on $\chi$, the $O_3$ mass mixing ratio ($kg\,kg^{-1}$), $T$, the temperature ($K$), and $\Phi$, the column $O_3$ above the point under consideration ($kg\,m^{-2}$).

Equation (7) is expanded to first order in a Taylor Series as follows:

$$\frac{\partial C_{O_3}}{\partial t} = \frac{d\chi}{dt} = (P-L)_0 + \overbrace{\left.\frac{\partial(P-L)}{\partial\chi}\right|_0 (\chi - \chi_0)}^{\mathbf{a}} +$$
$$\overbrace{\left.\frac{\partial(P-L)}{\partial T}\right|_0 (T - T_0)}^{\mathbf{b}} + \overbrace{\left.\frac{\partial(P-L)}{\partial\Phi}\right|_0 (\Phi - \Phi_0)}^{\mathbf{c}} \tag{8}$$



The second term in the expansion represents variations in the local $O_3$ amount (a), the third represents temperature effects (b) and the last term, called radiation term, accounts for the influence of non-local $O_3$ on the amount of solar radiation reaching the level under consideration (c). Specific terms in this equation (represented with the subscript 0) are coefficients applicable at the equilibrium state. In COPCAT these coefficients are obtained at equilibrium from the TOMCAT/SLIMCAT box model (Chipperfield, 2006). These terms are presented as functions of 24 latitudes, 24 model vertical levels and 12 months.

Heterogeneous processes describing the polar stratospheric chemistry are non-linear and depend on the three-dimensional structure of the atmosphere. COPCAT includes complete heterogeneous processes in their coefficients, considering heterogeneous and gas-phase chemistry to be consistent when applied in this linear $O_3$ parameterization. This kind of parameterization is in better agreement with the current state of knowledge of stratospheric heterogeneous chemistry than previous schemes (Monge-Sanz et al., 2011). For further description of the approach, the reader is referred to Monge-Sanz et al. (2011).

### 2.2.6 Online natural emissions

Natural emissions of gaseous pollutants include biogenic emissions, soil emissions, emissions from lightning, and emissions from oceans and volcanoes. Currently, soil and oceanic emissions in the model are prescribed as described in Sect. 3.1 and emission from lightning and volcanoes are not considered. Only biogenic emissions, which strongly depend on meteorological fields and vegetation cover, are calculated online. They are computed using the Model of Emissions of Gases and Aerosols from Nature version 2.04 (MEGANv2; Guenther et al., 2006). MEGAN is able to estimate the net emission rate of gases and aerosols from terrestrial ecosystems into the above-canopy atmosphere. MEGAN canopy-scale emission factors differ from most other biogenic emission models, which use leaf-scale emission factors, and cover more than 130 Non-Methane Volatile Organic Compounds (NMVOCs). All the MEGAN NMVOCs are speciated following the CB05 chemical mechanism; thus, emissions for isoprene, lumped terpenes, methanol, nitrogen monoxide, acetaldehyde, ethanol, formaldehyde, higher aldehydes, toluene, carbon monoxide, ethane, ethene, paraffin carbon bond, and olefin carbon bond are considered within the model. Biogenic emissions are computed every hour to account for evolving meteorological changes in solar radiation and surface temperature. Thus, weather- driving variables considered are temperature at 2 m and incoming short wave radiation at surface.

Figure S1 in the supplementary information shows the modeled emission for isoprene and terpenes for January and July 2004, and Table 2 lists the global annual emissions for isoprene, monoterpenes and other important NMVOCs. Biogenic isoprene emissions used in this study amount 683.16 Tg/year. While other global models have lower estimates (Huijnen et al., 2010; Horowitz et al., 2003; Emmons et al., 2010), MEGAN isoprene emissions typically range from about 500 to 750 Tg/year (Guenther et al., 2006). These estimates largely depend on the assumed land cover, emission factors, and meteorological parameters Therefore, the emission uncertainties and their impacts upon surface $O_3$ are associated with uncertainties in these inputs. Ashworth et al. (2010) obtained emission reductions of 3% and 7% when using daily and monthly meteorological data, respectively, instead of hourly data, with reductions reaching up to 55% in some locations. Marais et al. (2014) performed several sensitivity model runs to study the impact of different model input and settings on isoprene estimates that resulted in differences of up to $\pm$ 17% compared to a baseline. In our study, weather inputs are based on previous day 24h averages and data of the hour of interest.



## 3  Model setup

The model is set up as global with a horizontal grid spacing of $1.4° x 1°$ and 64 vertical layers up to 1 hPa. The dynamics fundamental time step is set to 180s and the chemistry processes are solved every 4 fundamental time steps. The radiation, photolysis scheme and biogenic emissions are computed every hour. We use NCEP/Final Analyzes (FNL) as initial conditions for the meteorological driver, and we reinitialize the meteorology every 24 h to reproduce the observed transport. We performed a spin-up of 1-year using initial chemistry conditions from the global atmospheric model MOZART-4 Emmons et al. (2010) prior to the 2004 annual cycle simulation that is evaluated in the present study. Table 1 describes the main configuration of the model. The feedback between chemistry and meteorology is not considered in this study.

### 3.1  Emissions

The global emissions used in this study are based on the Atmospheric Chemistry and Climate Model Intercomparison Project (ACCMIP; Lamarque et al., 2013), which includes emissions from anthropogenic and biomass burning sources at $0.5° x 0.5°$ horizontal resolution (Lamarque et al., 2010). Note that the 2004 emissions are derived from a linear interpolation between years 2000 and 2010. Therefore specific events occurred during 2004 (e.g., large summer wildfires in Alaska and Canada) are not described. The ACCMIP inventory is a combination of several existing regional and global inventories. The surface anthropogenic emissions are based on two historical emission inventories, namely RETRO (1960-2000; Schultz and Rast (2007)) and EDGAR-HYDE (1890-1990; Van Aardenne et al. (2005)) and monthly variations for biomass burning, ship and aircraft emissions are provided. One limitation is that land-based anthropogenic emissions have constant values for the entire year. Lamarque et al. (2010) presents a comparison of the annual total CO anthropogenic and biomass burning emissions (Tg(CO)/year) for different regional and global emission inventories for year 2000 (see Table 5 of this paper). Note that ACCMIP global CO anthropogenic emissions are significantly higher (610.5 Tg CO/year) than other emissions inventories (e.g. RETRO with 476 Tg CO/year, EDGAR-HYDE with 548 Tg CO/year, and GAINS with 542 Tg CO/year).

Ocean and soil natural emissions are based on the POET (Granier et al., 2005) global inventory. Lightning and volcano emissions are not considered in this simulation. Biogenic emissions are computed using MEGANv2 model as described in Sec. 2.2.6. NO emissions for January and July 2004 are shown in Figure S1 in the supplementary information and yearly totals for anthropogenic and biomass burning are summarized in Table 2.

To account for the sub-grid scale vertical diffusion within the planetary boundary layer (PBL) all the land-based anthropogenic emissions are emitted in the first 500 m of the model, biomass burning emissions from forests in the first 1300 m, biomass burning emissions from grass in the first 200 m, ocean emissions on the first 30 m and shipping emissions on the first 500 m. The model does not include the attenuation of radiation due to aerosols in the photolysis scheme. Therefore, regions with strong biomass burning emissions may significantly overestimate chemical photolysis production.



## 4 Observational data

### 4.1 Surface concentration and wet deposition

For the evaluation of ground-level gas concentrations, we selected background stations having hourly data (Fig. 1) from the World Data Center for Greenhouse Gases (WDCGG; http://gaw.kishou.go.jp/wdcgg/), the European Monitoring and Evaluation Programme (EMEP; http://www.emep.int/), the Clean Air Status and Trends Network in USA (CASTNET; http://java.epa.gov/castnet/) and the Acid Deposition Monitoring Network in East Asia (EANET; http://www.eanet.asia/). For $O_3$, we used data from 41 WDCGG, 52 EMEP, 64 CASTNET and 11 EANET stations, covering Europe, United States, and a few locations in east Asia. We also selected 21 EMEP stations for $NO_2$, 10 EANET stations for $NO_x$ and 14 WDCGG stations for CO.

The simulated wet deposition of $HNO_3$ is also compared against observed nitrate ($HNO_3$ and aerosol nitrate) wet deposition, including 260 measurements from the National Atmospheric Deposition Program (NADP; http://nadp.sws.uiuc.edu/) network in North America, 51 from the EMEP network in Europe and 28 from EANET in East Asia.

### 4.2 Vertical structure: ozonesondes and MOZAIC

The surface evaluation is complemented with an assessment of the vertical structure of $O_3$ using ozonosondes from the World Ozone and Ultraviolet Radiation Data Center ozonosonde network (WOUDC;http://www.woudc.org/), the Global Monitoring Division (GMD; ftp://ftp.cmdl.noaa.gov/ozwv/ozone/) and the Southern Hemisphere ADditional OZonesondes (SHADOZ; http://croc.gsfc.nasa.gov/shadoz/; Thompson et al., 2003a, b). Most stations provide between 4 and 12 profiles per month each year with a precision of $\pm$ 3-8 % in the troposphere (Tilmes et al., 2012). We followed the methodology of Tilmes et al. (2012) for the selection and treatment of the measurements. Table 3 lists the locations and the number of available measurements per season of the 39 ozonesonde stations used (also displayed in Fig. 1), as well as the regions where stations with similar $O_3$ profiles were aggregated).

Additional observations considered in this study are CO vertical profiles from Measurement of Ozone, Water Vapor, Carbon Monoxide, Nitrogen Oxide by Airbus In-Service Aircraft (MOZAIC; http://http://www.iagos.fr). Based on the availability of data, we selected 14 airports (displayed in Fig. 1) covering different regions of the world during 2004. The number of vertical profiles available per season are provided in Table 4.

Nitric oxide ($NO_x$), peroxyacetyl nitrate (PAN) and acid nitric ($HNO_3$) vertical profiles are used from two different measurement campaigns: TOPSE (Atlas et al., 2003; Emmons et al., 2003) and TRACE-P (Jacob et al., 2003). Tropospheric data from these two previous campaigns were gridded onto global maps with resolution 5°x5° x1km, forming data composites of important chemical species in order to provide a picture of the global distributions (Emmons et al., 2000).

When running an AQM model, it is preferable to compare the model output with an observational database from the same year as the model simulation. Nevertheless, in our case, there are insufficient global observations to achieve this goal for any full year. Hence, in this model evaluation, all the observations are for 2004, except for the vertical profiles obtained from measurement campaigns. Hence, in this study, model output from selected regions are compared with this campaign from the same regions regardless of the year of the measurements. In addition, it is valuable to compare the same regions for different





species which allows identification of systematic differences between the model results and observations (Emmons et al., 2000). Details of these campaigns describing their geographical region and period are described in Table 5, and the location displayed in Fig. 1 (right panel).

### 4.3 Satellite data

Modelled tropospheric $NO_2$ columns are compared with SCanning Imaging Absorption spectroMeter for Atmospheric CHartographY (SCIAMACHY, http://www.sciamachy.org/) satellite data. SCIAMACHY (on board of ENVISAT, which was operational from March 2002 to April 2012) is a passive remote sensing spectrometer measuring backscattered, reflected, transmitted or emitted radiation from the atmosphere and Earth's surface with a wavelength range between 240-2380 nm. The SCIAMACHY instrument has a spatial resolution of typically 60 x 30 km$^2$, and has three different viewing geometries: nadir,

limb, and sun/moon occultation. Alternating nadir and limb views, global coverage is achieved in six days.

$NO_2$ daily data was obtained from the Institute of Environmental Physics, the University of Bremen (http://www.iup.uni-bremen.de/doas/scia_no2_data_tropos.htm), based on Version 3.0 data product (Hilboll et al., 2013). This dataset is an improved extension of the data presented in Richter et al. (2005). Validation of the data product was performed in several studies (e.g., (Petritoli et al., 2004), (Heue et al., 2005)). We used daily satellite overpasses of cloud-free ($<20\%$ cloud fraction)

tropospheric vertical column densities (VCDtrop $NO_2$) from SCIAMACHY measurements using the limb/nadir matching approach, whose total uncertainty is estimated to vary between 35 and 60% in heavily polluted cases and $>100\%$ in clean scenarios (Boersma et al. (2004)).

Additionally, CO mixing ratios at 800 and 500 hPa were evaluated with the Measurement of Pollution in the Troposphere (MOPITT: http://www2.acd.ucar.edu/mopitt) instrument retrievals. The MOPITT, aboard the NASA EOS-Terra satellite, is a

gas filter radiometer and measures thermal infrared (near 4.7 $\mu m$) and near-infrared (near 2.3 $\mu m$) radiation, only during clear-sky conditions, with a ground footprint of about 22 km x 22 km. We used the MOPITT Version 5 (V5) Level 2 data product, which provides daily CO mixing ratios. MOPITT CO mixing ratios have been validated with in situ CO profiles measured from numerous NOAA/ESRL aircraft profiles in Deeter et al. (2013), and they were found to be positively biased by about 1% and highly correlated (r = 0.98) at the surface level.

## 5  Model evaluation

This section presents the evaluation of relevant trace gases from the NMMB/BSC-CTM using the observations described in the previous section. It also compares the results with other modeling studies available in the literature. For the surface-level comparison, three-hourly averages from the observations and model are used to compute daily $O_3$ averages and calculate the statistical measures defined in section 1 in the supplemental material. Ground-monitoring stations were selected with a

maximum altitude of 1000 meters. In the case of ozonesondes and MOZAIC the comparison is made only when vertical profile observations are available, i.e., the data from the model and the observations are collocated/simultaneous. Similar criteria is used in the case of MOPITT and SCIAMACHY for $NO_2$. Moreover, averaging kernels for CO are accounted to represent



the observational sensitivity at different pressure levels. When computing the modelled tropospheric columns of $NO_2$ the tropopause was assumed to be fixed at 100 hPa in the tropics and 250 hPa in the extratropics.

Similarly, when comparing model data with data composites from aircraft campaigns the same period of the year at the same location is selected and mapped into the same grid resolution, 5° x 5° x 1km, before the comparison is made. For some species, the model evaluation is given per seasons: DJF for December-January-February, MAM for March-April-May, JJA for June-July-August and SON for September-October-November.

### 5.1 Hydroxyl Radical (OH)

One of the means for characterizing the general properties of an AQM is through its ability to simulate OH oxidation. OH is the main oxidant in the troposphere and is responsible for the removal of many compounds, thereby controlling their atmospheric abundance and lifetime. OH is mostly found in the tropical lower and mid troposphere with a strong dependence on the levels of ultraviolet radiation and water vapour. Tropospheric OH formation is mainly due to $O_3$ photolysis, dominated by the tropics. Also, OH is directly connected to the chemistry of $O_3$ production since the initial reactions of $O_3$ formation (VOC+OH and CO+OH) are driven by OH. Hence, $O_3$ production rates depend on the sources and sinks of odd hydrogen radicals. Primary OH formation also includes the photolysis of HCHO and secondary VOC.

The tropospheric mean (air mass weighted) OH derived by the model is 11.5 molec $10^5$ cm$^{-3}$, assuming a tropospheric domain ranging from 200 hPa to the surface. (Note that previous studies suggest that the estimation of the mean OH does not depend on the definition of the tropopause (Voulgarakis et al., 2013).) This value is in good agreement with other studies, e.g., Voulgarakis et al. (2013) where the mean OH concentration from 14 models for 2000 was estimated to be $11.1\pm1.8$ $10^5$ molec cm$^{-3}$; Spivakovsky et al. (2000) with 11.6 $10^5$ molec cm$^{-3}$, and Prinn et al. (2001) with $9.4 \pm 0.13$ $10^5$ molec cm$^{-3}$.

The zonal mean OH concentrations for January, April, July and October 2004 are shown in Fig 2. Seasonal differences reflect the impact of water vapor concentration and stratospheric $O_3$ column upon incident ultraviolet (UV) radiation (Spivakovsky et al., 2000; Lelieveld et al., 2002). The highest OH concentrations arise in the tropics throughout the year. In northern midlatitudes, the highest OH concentrations are found during summer in the lower to middle troposphere. The latitudinal and seasonal variations are similar to the climatological mean in Spivakovsky et al. (2000), particularly the lower values in the extratropics. Peak concentrations are slightly larger compared to this climatology and other studies (e.g., Horowitz et al., 2003; Huijnen et al., 2010). During January and October the peaks appear in the southern tropics between 700-1000 hPa and 800-1000 hPa, respectively. The peak in April and July is found in the northern tropics between 800-1000 hPa and 700-1000 hPa, respectively. The larger oxidizing capacity compared to other studies could be due to the lack of aerosols in our simulation, which may overestimate photolysis rates in polluted regions.

### 5.2 Carbon monoxide (CO)

CO is one of the most important trace gases in the troposphere exerting a significant influence upon the concentration of oxidants such as OH and $O_3$ (Wotawa et al., 2001). Main sources of CO in the troposphere are the photochemical production from



the oxidation of hydrocarbons and direct emissions, mainly fossil fuel combustion, biomass burning and biogenic emissions. CO main loss is by reaction with OH, which occurs primarily in the tropics, but also in the extratropics.

In the northern extratropics, the elevated CO concentrations are dominated by anthropogenic emissions and precursor hydrocarbons, which leads to a net CO export to the tropics (Shindell et al., 2006; Bergamaschi et al., 2000). Although most biomass burning occurs in the tropics, gases and aerosols emitted from large wildfires can be transported to the southern extratropics, where emissions and chemical production are lower. Moreover, due to the strong convection, enhanced by forest fire activity, emissions can reach the upper troposphere and the lower stratosphere (Jost et al., 2004; Cammas et al., 2009). CO has a chemical lifetime of a few months ($\sim$1-3), and therefore it is a useful tracer for evaluating transport processes in the model. It is important to keep in mind that despite large Alaskan and Canadian wildfires occurred during the summer, globally 2004 had lower CO concentrations than other years during the decade (Elguindi et al., 2010).

An analysis of the CO burden in different regions is presented in Table 6. The global and annual mean burden of CO for 2004 is 399.03 Tg, with higher abundances in the tropics (229.43 Tg CO), followed by the Northern Extratropics (101.71 Tg CO), and the Southern Extratropics (67.88 Tg CO). Other model estimates of the CO burden (Horowitz et al. (2003);Huijnen et al. (2010)) are also shown in Table 6. Our estimates are higher ($\sim$ 46-48 Tg CO) in comparison with these studies in all regions. The largest absolute difference appears in the tropics where the NMMB/BSC-CTM predicts $\sim$ 30-40 Tg CO more than these studies, even though OH is also overestimated. The main sources of CO in the tropics are from biomass burning, biogenic emissions and anthropogenic direct emissions of CO.

We performed tests comparing the annual mean burden of tropospheric CO with and without biomass burning emissions in the model. Neglecting biomass burning emissions only reduced 7% of the tropospheric CO annual mean burden. Therefore, other factors should explain our higher CO burden. On the one side, biogenic emissions are computed online every hour in order to account for evolving meteorological changes, such as solar radiation and surface temperature (see section 2.2.6). Also this simulation neglects the attenuation of radiation due to aerosols, which may produce an overestimation of VOCs biogenic emissions and the derived CO.

The CO anthropogenic emissions used in this study (610.5 Tg/year) are are also higher than those in other inventories (see 3.1). The dry deposition of CO is significantly weaker in the NMMB/BSC-CTM (24 Tg CO) than the global model TM5 (184 Tg CO) and the study of Bergamaschi et al. (2000) (292-308 Tg CO). By contrast, other global models such as MOZART-2 have significantly lower dry deposition (2 Tg CO) and the study of Wesely and Hicks (2000) suggests that CO and other relatively inert substances are deposited very slowly. Clearly, there are major uncertainties in the sources and sinks of CO that could be responsible for modeled CO differences.

Fig 3 shows the time series of CO daily mean concentration over 14 ground-monitoring stations from the WDCGG database (primarily in the northern mid-latitudes, but with a few of them in the tropics and southern mid-latitudes). The solid red line and the solid black line represent, respectively, the average of observations and the model simulation. Bars show the 25th-75th quartile interval of all observations (orange) and the model simulation (grey). The model is in good agreement with the CO field in the surface layer. However, the model is not able to fully capture the seasonal CO variability, with a slight underestimation during cold months and overestimation during warm months. Such a model limitation could be explained by the fact that most





of the stations are closer to anthropogenic polluted areas, where its concentration is primarily determined by local emissions, and the CO land-based anthropogenic emissions inventory does not have any seasonal variation in this study (see Sec.3.1).

Fig. 4 shows the CO mean bias (MB), correlation and root mean square error for all rural WDCGG stations. The model has a negative MB over stations in Europe and Japan and a positive bias in stations in Canada and Africa, where the correlations are low. The negative bias for several of the northern mid-latitude stations indicates that the higher CO burden found in our model compared to other models in these areas is a feature mainly driven by free tropospheric abundances. Higher correlations are found in northern regions of Europe, southern Africa and eastern Asian countries. Correlation in Canadian stations is between 0.3-0.5. In most of the stations, RMSE is found to be less than 60-40 $\mu$g $m^{-3}$; only 4 stations have an RMSE higher than 60 $\mu$g $m^{-3}$.

Additionally, the model was compared with the seasonally averaged vertical profiles of CO from MOZAIC aircraft observations for 2004 in Figs. 5 and S2 in the supplementary information from selected airports: Frankfurt, Beijing, Atlanta, Portland, Abu Zabi and Niamey. The comparison is made only when observations are available; i.e., the same data from the model and the observations are used. Measurements are represented by the solid red line and the model simulation by the solid black line. To understand the variability of the data, standard deviation is plotted in each vertical layer for both model and observations. It is important to note that the number of flights is significantly different between the different airports; therefore, not all comparisons are statistically robust. In addition, note that the scale for Beijing is different (0-1000 ppb) from the others stations (0-400 ppb).

The model captures well the vertical profiles during the first part of the year, with higher biases during the warm months. Generally it overestimates CO from the middle to the upper troposphere in most of the stations throughout the year. Over Frankfurt, the model is in good agreement with the observation during the entire year, despite a slight underestimation during MAM and overestimation during SON in the middle troposphere.

For Beijing, one of the most polluted cities in the world, the model shows a clear tendency to underestimate CO in the lower atmosphere (below 600 hPa). This is probably due to an underestimation in the CO anthropogenic emissions. Most of the AQMs seem to be unable to capture the extreme growth of anthropogenic emissions in China (Akimoto, 2003; Turquety et al., 2008). Over Atlanta, the model performs much better during the winter and spring along the troposphere but positive biases ($\sim$ 20-25 ppb) are seen during the summer and autumn. Regions with biomass burning and biogenic influence, such as Abu Dhabi and Niamey, show a significant overestimation during warm months throughout the tropospheric column. During winter, Stein et al. (2014) also obtain an underestimation of CO vertical profiles in airports located in the Northern Hemisphere (NH).

To complete this CO evaluation, seasonal averages are compared with data from the MOPITT instrument at 800 hPa and 500 hPa in Figs. 6 and S3 in the supplementary information, respectively. At 800 hPa, largest differences are seen during boreal winter and spring, where the model clearly overestimates in the tropics and underestimates in the north extratropics and north of Africa. The negative bias during winter ($\sim$ 10-35 ppb) in the NH could further be explained by the lack of a seasonal cycle in anthropogenic emissions. However, the underestimation during NH winter, which appears most state-of-the-art AQMs could also be originated from an underestimation of CO emissions (Stein et al., 2014).



There are significant positive biases over west-central Africa and also over western South America, Indonesia and the surrounding Pacific and Indian oceans during the dry season. Sources of CO over west-central Africa are mainly from biomass burning and biogenic emissions. Uncertainties in the emission inventories have probably contributed to the CO overestimation for these regions. Due to the long-range transport of CO, higher CO concentrations are seen throughout all the year over the

tropics and are extended over some parts of the extratropics from June to November. Hence, during JJA and SON the model overestimates CO concentrations in most places including south and central EU and USA ($\sim$ 10-25 ppb).

At 500 hPa, the model presents similar results, with a clear underestimation in the north extratropics and overestimation in the tropics and southern latitudes. Overestimated emissions in Africa or Asia above the PBL can lead to this positive bias in the middle of the troposphere.

Naik et al. (2013) presents an annual average bias of multi-model (17 global models) mean CO for 2000 against average 2000-2006 MOPITT CO at 500 hPa. These models used the same anthropogenic and biomass burning emissions as our study, and a priori and averaging kernels are taken into account for each model before computing biases. The biases in the tropics and extra tropics are similar to those presented here. Hence, these biases might be related to discrepancies in anthropogenic and biomass burning emission inventories, where the magnitude, and perhaps location of emission is not completely understood

or correctly modelled. Naik et al. (2013) discussed a too high OH concentration possibly leading to the northern mid-latitude underestimates of CO, which is also a possibility in our case, given the high OH concentrations that the model shows compared to other models. Numerous studies show that the variability in simulated CO among AQMs is large, and uncertainties are diverse including emission's inventories and injection height (Elguindi et al., 2010; Shindell et al., 2006; Prather et al., 2001). A detailed evaluation of MOPITT V4 CO retrievals between 2002-2007 with in situ measurements shows a bias of about -6%

at 400 hPa (Deeter et al., 2010). However, this bias is not able to explain the model biases that vary in sign and magnitude between different global regions. Stein et al. (2014) suggests that the persistent negative bias in northern mid-latitude CO in models is most likely due to a combination of too low road traffic emissions and dry deposition errors.

### 5.3 Nitrogen compounds

The $NO_x(= NO_2 + NO)$ family is one of the key players in the formation of $O_3$ in the troposphere, and during pollution

episodes it causes photochemical smog and contributes to acid rain. It has a relatively short lifetime; consequently, it is generally restricted to emission sources, both natural and anthropogenic (mainly fossil fuel combustion). The seasonal cycle of $NO_x$ near the surface is controlled by the seasonality of anthropogenic emissions (especially in the northern hemisphere) and biomass burning emissions (especially in the tropics and the southern Hemisphere). As a result, $NO_x$ is more sensitive to errors in emissions than other pollutants, and errors in $NO_x$ emissions can change $NO_x$ concentrations even more drastically (Miyazaki

et al., 2012).

Figure 7 shows the time series of $NO_2$ and $NO_x$ daily mean surface concentrations over 21 and 10 ground-monitoring stations from the EMEP and EANET networks, respectively. In both cases, the model is able to successfully reproduce the seasonal cycle of $NO_2$ and $NO_x$. However, a positive bias is found during the summertime for $NO_2$ in Europe (Fig. 7 top panel). Such a result could be explained by the limitation on the anthropogenic emissions that are constant during the entire





year. Because of that, the model cannot reproduce the decrease in anthropogenic emissions during the summertime, which leads to overestimated concentrations.

Daily profiles show that the modeled $NO_2$ tends to be too high during nighttime (not shown). This result may be due to the lack of the heterogeneous formation of $HNO_3$ through $N_2O_5$ hydrolysis, an important sink of $NO_2$ at night. In addition, the

model does not consider secondary aerosol formation for the present study, which might result in an atmosphere that is too oxidising (overestimation of OH radicals). In combination with the nocturnal chemistry this may lead to an accumulation of $NO_2$ in the surface layers. However, a slight underestimation is observed between 9-18 UTC. Looking at the annual time series of $NO_x$ in the Asian network (Fig. 7 bottom), it is observed that the model does not reproduce $NO_x$ values, with a sizeable negative bias during the summer. This underestimation could be attributed to an underestimation in the emission inventories,

which do not capture the extreme increase of anthropogenic emissions over Asia during the last decade (Akimoto, 2003; Richter et al., 2005), as was the case for CO.

Concerning the spatial statistics (see Fig. 8 ), the model's prediction capabilities are lower in some regions such as the Iberian Peninsula and most of the stations in Japan, showing poor correlations. The best performance is seen in central EU and stations in Japan that are not in the main island. In general there is a negative bias in most of the stations for these two regions.

The comparison of modelled and observed vertical profiles of $NO_X$, $HNO_3$ and PAN are presented in Fig. 9 for several regions over US, China, Hawaii and Japan (see Table 5). As explained in Sec. 4.2, the observed vertical profiles do not correspond to the simulated year (see Table 5 for more detail), but the qualitative patterns can provide insights on the model capability to reproduce the chemistry involved. Fig. 9 (first column) shows that vertical profiles of $NO_x$ are in very good agreement with the observed values. The model has a tendency to overestimate $NO_X$ concentrations near the surface; it is

likely that $NO_X$ emissions used in this study are higher than the real emissions during the campaigns' periods. Another reason for these higher values over island locations (Japan and Hawaii) could be that emissions are spread throughout the entire low resolution model grid box while the measurements were taken in the cleaner marine boundary layer. In the middle and upper troposphere, the model produces the concentrations well, with a slight underestimation in most of the locations. Note that $NO_x$ lightning emissions are not included in this simulation, which may explain part of this underestimation, particularly in the

upper troposphere.

PAN is the main tropospheric reservoir species for $NO_x$ with important implications for the tropospheric $O_3$ production and the main atmospheric oxidant, OH (Singh and Hanst, 1981). PAN is mainly formed in the boundary layer by oxidation of NMVOCs in the presence of $NO_x$. NMVOCs and $NO_x$ have both natural and anthropogenic sources. Rapid convection can transport PAN to the middle and upper troposphere and enables the long-range transport of $NO_x$ away from the urban and

polluted areas, where it can produce $O_3$ and OH remotely.

Some features of the vertical profiles are well-captured by the model, although it significantly overestimates PAN concentrations (see Fig. 9, second column). We find overestimations from the surface to the middle atmosphere in Japan, China, Boulder and Churchill, which are possibly explained by an overestimation of biogenic and anthropogenic $NO_x$ surface emissions in this area at surface-level. Another possibility for this overestimation is a too long lifetime for PAN. At most sites, PAN model

concentrations tend to increase with altitude, reaching maximum mixing ratios at about 6km, from where they progressively



decrease. This behaviour explains the long thermal decomposition time of PAN (lifetime of approximately a month) and the slow loss by photolysis in the cold middle-upper troposphere. Fischer et al. (2014) analyse the sensitivity of PAN to different emission types, showing that most of the northern hemisphere and Japan are more sensitive to anthropogenic emissions, while the southern hemisphere and the west coast of the USA are more sensitive to biogenic emissions, both contributing to 70-90%

of the PAN concentrations.

$HNO_3$ is mainly produced by the reactions of $NO_2$ with OH and by the hydrolysis of $N_2O_5$ on aerosols (we do not account for this reaction in this simulation), and then it is removed by wet and dry deposition. $HNO_3$ is the main sink of $NO_x$ chemistry. In general, the modelled and observed nitric acid concentrations are in good agreement throughout the troposphere, although the model reveals a tendency to overestimate $HNO_3$ concentrations that is even more pronounced in US regions. In the regions

of Hawaii, Japan and China the model overestimates $HNO_3$ in the lower-middle troposphere (up to 5km) and underestimates it in the upper troposphere (above 6km). Overestimation of $HNO_3$ in the troposphere is a common problem in global models (Hauglustaine et al., 1998; Bey et al., 2001; Park et al., 2004; Folberth et al., 2006). $HNO_3$ concentrations are highly sensitive to the parameterization of wet deposition. One possible reason for this overestimation is that the scavenging from convective precipitation is underestimated.

Figure S4 in the supplementary information presents the wet deposition fluxes of $HNO_3$ in comparison with nitrate observations for three different networks located in Europe, USA and Asia. Satisfactory agreement is found in the $HNO_3$ wet deposition fluxes with correlations of 0.63 in Europe, 0.80 in USA and 0.52 in Asia. There is a tendency to underestimate in most of the stations, principally in Asia and Europe. Part of this underestimation is because the comparison is between nitric acid (gas) and nitrate (nitric acid + particulate nitrate) wet depositions. However, this tendency to underestimate is consistent

with the higher values of $HNO_3$ observed at the lower and middle troposphere.

Seasonal averages of vertical tropospheric columns (VTC) of $NO_2$ are compared with SCIAMACHY satellite data in Fig.10. The model is in good agreement with the observations, capturing the high $NO_2$ values over the most polluted regions, such as Europe, USA and Eastern Asia. The phase in the seasonal cycle of the $NO_2$ columns is performed well by the model. During the entire year, the model tends to underestimate $NO_2$ VTCs in big cities, especially during the colder months, and overestimate

them in rural regions. The largest discrepancies are seen in eastern China, which suggests an underestimation of emissions regionally. The biomass burning cycle is captured remarkably well, with higher $NO_2$ VTC in central Africa during DJF and in South America in JJA. Over the sea, the model is in good agreement with SCIAMACHY, showing only small differences ($\pm$ 0.5 $1e^{15}$ molec/cm$^2$).

### 5.4 Ozone ($O_3$)

Ozone is one of the central species that drive tropospheric chemistry, and for that reason it is essential that a model reproduces the spatial and temporal concentrations of $O_3$ well, both at the surface and across the troposphere and stratosphere. $O_3$ found in the troposphere is originated from in situ photochemical production and from intrusions of $O_3$ from the stratosphere. $O_3$ photochemical production in the troposphere involves oxidation of CO and hydrocarbons in the presence of $NO_x$ and sunlight.



In rural areas, CO and CH$_4$ are the most important species being oxidized in the O$_3$ formation. However, in polluted areas, short-lived NMVOCs (e.g. HCHO) are present in high concentrations and are the most important species.

The simulated global burden of tropospheric O$_3$ is shown in Table 7. In the troposphere, O$_3$ chemical sources and sinks are dominated by the tropics, where high concentrations are found (171.60 Tg O$_3$). Low concentrations are predicted in the northern extratropics (101.56 Tg O$_3$) and especially the Southern Extratropics (75.41 Tg O$_3$), where precursors are not present in high amounts. Similar results are found from other global models, such as MOZART-2 (Horowitz et al., 2003) and TM5 (Huijnen et al., 2010). In general, MOZART-2 has a higher and TM5 a lower annual mean burden of O$_3$ than the NMMB/BSC-CTM. The annual mean O$_3$ burden predicted by our model in the southern extratropics is higher (10-14 Tg O$_3$) than the other two models. Higher CO concentrations in the southern hemisphere (see Table 6) might lead to excessive production of O$_3$ in this area. In addition, the global tropospheric O$_3$ burden in our model (348 Tg O$_3$) is in good agreement with the C-IFS global model (Flemming et al., 2015) and the two multimodel ensemble means of 25 (Stevenson et al., 2006) and 15 (Young et al., 2013) state-of-the-art atmospheric chemistry global models.

According to our calculations, 1209 Tg O$_3$ are removed from the troposphere by dry deposition at the surface. This quantity is higher in comparison with the global models TM5 (829 Tg O$_3$) and MOZART-2 (857 Tg O$_3$). In this sense, the model is in good agreement with the global model LMDz-INCA (1261 Tg O$_3$) and with the multimodel ensemble study by Stevenson et al. (2006) (1003 $\pm$ 200 Tg O$_3$). The net stratospheric input, Stratosphere-Troposphere Exchange (STE), annual rate of the model (384 Tg O$_3$) is also shown in Table 7. The model's STE is in good agreement with other modelling studies, especially with the multimodel ensemble in Stevenson et al. (2006) (552 $\pm$ 168 Tg O$_3$).

Fig.11 shows the time series of O$_3$ daily mean concentration averaged over all available monitoring sites (from top to bottom, WDCGG, CASTNET, EMEP and EANET) over the entire simulation period. The solid red line and solid black line represent the average of observations and the model, respectively. Bars show the 25th-75th quartile interval of all observations (orange) and model simulation (grey). As illustrated in Fig.11, there is an overall good performance although there are significant positive bias from May to October in the US and Japan. The seasonal cycle of O$_3$ from the model agrees well with the observations, showing the highest concentrations during July-August and the lowest concentrations during Nov-Dec over all stations. Although, the model captures the seasonal O$_3$ variability along this period, there is a tendency to overestimate concentrations during the warmer months, i.e. May-September. This positive bias is significantly higher in the US, where the overestimation occurs all day long (10-20 $\mu$g m$^{-3}$). Over Europe, the overestimation of O$_3$ levels during summer is lower than in the other regions. Over East Asia the model captures reasonably well the peaks in April and May, although a positive bias is seen during the rest of the year. In this area, concentrations during cold months are overestimated, in contrast to Europe where the model concentrations agree with the observations. Overall the observational networks show a reduction of O$_3$ concentrations from May-June, but the model has a tendency to simulate an annual cycle with higher concentrations until July. Moreover, a recent study by Val Martin et al. (2014) explains the importance of the dry deposition in controlling surface O$_3$. Val Martin et al. (2014) shows that accurate dry deposition processes can reduce the summertime surface O$_3$ bias from 30 ppb to 14 ppb and from 13 ppb to 5 ppb over eastern U.S. and Europe, respectively. Thus, part of this positive bias could be related



to the dry deposition processes included in our model. Further investigation is required to understand model behaviour during this period.

Fig.12 displays the spatial statistics for $O_3$ over all in-situ monitoring sites using daily mean data. Areas without emissions, such as the south pole and isolated islands in the tropics, have small mean biases and errors and good correlations ($>0.80$). In polluted areas, a good performance is observed in the US midlands, and parts of central and southern Europe ($0.60< r<0.80$ and RMSE $<20$ $\mu$g $m^{-3}$). Large errors are seen in northwestern and southern US and Northern Europe. Although, large errors are seen in all the stations over Japan, the two stations farthest from the main island show high correlation ($r> 0.7$).

In order to assess the vertical distribution of $O_3$, the model results are compared with available ozonesondes. The seasonal vertical profiles of $O_3$ for both the model and observations are compared in Figs.13 and S5 in the supplementary information for the period of study (see Table 3 and Fig. 1 for more details). The comparison is made only when ozonesonde observations are available. Fig.13 and S5 from the supplementary information show (from top to bottom) four panels: DJF, MAM, JJA and SON for each region. Measurements are represented by the solid red line and the model results by the solid black line. The variability of the data is shown in the form of standard deviation for both the model and observations.

The magnitude and vertical profile of $O_3$ are in good agreement with the observations. However, the model shows a positive bias of $\sim$ 5-20 ppb along the troposphere in most of the regions during the entire year. As shown in Sec. 5.2 there is a significant overestimation of CO, especially in the free troposphere for some regions, which may account for the positive $O_3$ biases, although the CO overestimation mostly occurs in the tropics where $O_3$ biases are not so large. Another reason for this result could be that anthropogenic aerosols and secondary aerosol formation are neglected in this simulation, leading to a higher $O_3$ formation in regions with more precursors. However, this should have more localised effects and therefore it cannot fully explain the biases throughout the troposphere.

The vertical profile is in good agreement with the observations, with $O_3$ increasing from lower to higher tropospheric layers. In the lower-middle troposphere the model overestimates $O_3$ in regions with high emissions (Japan, Canada, USA and W.Europe), a feature that is more significant in DJF. In Western Europe and the US, this bias is reduced at the surface level. In tropical areas (Equator, NH tropical and W. Pacific) the model captures well the observed concentration and vertical structure of $O_3$ in the lower to middle troposphere. However, the model tends to overestimate the $O_3$ in the vicinity of the tropopause layer in these regions. At polar regions (NH and SH Polar) the model also presents a tendency to overestimate the vertical structure of $O_3$. $O_3$ in the tropopause layer is underestimated in the NH Polar case, and overestimated in SH Polar case.

Finally, statistics were computed to identify those areas where the errors are more important. Fig. 14, shows the mean $O_3$ bias (left), correlation (middle) and RMSE (right) of the model with respect to ozonesondes (data is averaged between 400 and 1000 hPa over the year 2004). As we have shown, the mean bias is positive for most stations (MB$<30\%$). Large RMSE are seen in northern high latitudes ($<50$ $\mu$g $m^{-3}$) and in two stations from the US. Europe and Japan present an RMSE around $30\mu$g $m^{-3}$ and the tropics and subtropics are regions with lower errors, i.e. RMSE below $30\mu$g $m^{-3}$. The highest correlations are seen in polar regions.



## 6  Conclusions

A new global chemical transport model, NMMB/BSC-CTM, has been presented. A comprehensive description has been provided for the different components of the gas-phase chemical module coupled online within the NMMB atmospheric driver. This model, which includes 51 chemical species and solves 156 reactions, simulates the global distributions of ozone and its

precursors, including CO, $NO_x$, and VOCs. The model simulation presented here is configured with a horizontal resolution of 1° x 1.4°, with 64 vertical layers and a top of the atmosphere at 1 hPa. Emissions from ACCMIP (Lamarque et al., 2010) are considered and include fossil fuel combustion, biofuel, biomass burning, soil and oceanic emissions. Biogenic emissions are calculated online with the MEGANv2.04 model (Guenther et al., 2006). In this simulation, aerosols are neglected, thus, no interaction between gas-phase and aerosol-phase is considered.

Modelled tropospheric ozone and related tracers have been evaluated for the year 2004 and compared with surface-monitoring stations, ozonesondes, satellite and aircraft campaigns.

The evaluation of OH concentrations shows a good agreement with previous studies (Spivakovsky et al., 2000; Voulgarakis et al., 2013). The peak concentrations of OH seen in April and July at northern latitudes are slightly higher than the climatological mean calculated in Spivakovsky et al. (2000). This may be possibly explained by the fact that anthropogenic aerosols

and secondary aerosol formation are negelcted in this simulation; hence, a higher oxidized atmosphere is obtained due to higher photolysis when aerosols are not present. However, overall, the widespread positive ozone biases identified seem to be responsible for the higher OH concentrations.

The global annual mean burden of CO (399 Tg) is higher than in other studies, with larger concentrations located in the tropics (229.43 Tg CO). The model is in relatively good agreement with CO observations at the surface, and shows negative

biases at stations over Europe and Japan, and positive biases in Canada and Africa. The largest correlations are found in northern Europe, southern Africa and eastern Asia.

Concerning the vertical structure of CO, the model presents a good performance during the DJF and MAM, and positive biases are seen during JJA for most of the stations. In general, the model overestimates CO from the middle to the upper troposphere in most of the stations throughout the year. Significant underestimation of CO is seen in Beijing below 600 hPa.

This result is similar to other evaluation studies, which indicates that emission inventories are not able to capture the extreme growth of anthropogenic emissions in China. The phase and amplitude of the seasonal cycles of CO at 800 and 500 hPa in NMMB/BSC-CTM and MOPITT are quite similar.

Overestimations of CO are mainly located over west-central Africa, western South America, Indonesia and the surrounding Pacific and Indian oceans during the dry season. At 800 hPa, a significant negative bias is observed over the northern latitudes

during winter. These results are most likely related to errors in anthropogenic and biomass burning emission inventories, where the magnitude and the location of emission are not correctly represented. In addition, CO production from VOCs biogenic emissions, calculated online and depending on meteorological variables such as radiation, might be overestimated too, due to the lack of aerosol attenuation of radiation.



Nitrogen oxide abundances are well simulated in almost all locations. Looking at the annual time series of $NO_2$ in Europe, the model captures the higher peaks during winter, although a positive bias is observed during summer. Nitrogen compounds are more sensitive to errors in emissions than other pollutants. We note that the emission inventory neglects seasonal variations for land-based anthropogenic emissions, and therefore we do not account for the potential reduction of $NO_X$ emissions during

summer. Over Asia, there is a negative bias of $NO_x$ from March to August, probably due to underestimated emissions in this area. Vertical profiles of $NO_X$ are in good agreement with the observed values, although there is some underestimation in the upper troposphere, possibly due to the lack of lightning $NO_x$ emissions. Vertical profiles of PAN and $HNO_3$ were also compared with observations. Some agreement is seen in these vertical profiles, although the model has a tendency to over estimate. $HNO_3$ wet deposition fluxes tend to be underestimated and are better captured in the US compared to Europe and

Asia.

The comparison with observed $NO_2$ VTC from SCIAMACHY shows that the model reproduces the seasonality and the spatial variability reasonably well, capturing higher $NO_2$ over the most polluted regions. However, the results show a tendency to underestimate $NO_2$ VTC in big cities, especially during DJF and SON, possibly due to a low bias in the $NO_X$ emissions. The biomass burning cycle is well captured by the model with higher $NO_2$ VTC in central Africa during DJF and in South

America in JJA.

The ozone burden is in good agreement with other estimates from state-of-the-art global atmospheric chemistry models. The ozone burden in the southern extratropics is higher in our model, suggesting that higher CO concentrations in the southern hemisphere could lead to excessive production of ozone in this area. It seems unlikely that the positive ozone biases are caused by too much STE. STE is in good agreement with other evaluation studies. In addition, STE has stronger effects in the upper

troposphere, hence, the related biases should increase with height, which is not the case in our simulations.

The surface $O_3$ results show a reasonable agreement with the observations, with significant positive biases from May to October in the regions of the US and Japan. Surface $O_3$ concentrations are very sensitive to the emissions; consequently, the variability of ozone concentrations can be enhanced by improving the spatio-temporal distribution of the ozone precursor emissions.

The model captures the spatial and seasonal variation in observed background tropospheric $O_3$ profiles with a positive bias of $\sim$ 5-20ppb along the troposphere in most of the regions during the whole year. The significant overestimation of CO especially in the free troposphere could be the reason for this positive ozone bias.

In summary, NMMB/BSC-CTM provides a good overall simulation of the main species involved in tropospheric chemistry, although with some caveats that we have highlighted here. Future versions of the model will aim to address problems identified

in this study and will include the effect of aerosols in the system.

## 7   Code Availability

Copies of the code are readily available upon request from the corresponding authors.



*Acknowledgements.* The authors wish to thank WOUDC, GAW, EMEP, WDCGG, CASTNET-EPA, NADP and EANET for the provision of measurement stations. Also, thanks go to the free use of the MOPITT CO data obtained from the NASA Langley Research Center Atmospheric Science Data Center. SCIAMACHY radiances have been provided by ESA. This work is funded by grants CGL2013-46736-R, Supercomputación and e-ciencia Project (CSD2007-0050) from the Consolider-Ingenio 2010 program of the Spanish Ministry of Economy and Competitiveness. Further support was provided by the SEV-2011-00067 grant of the Severo Ochoa Program, awarded by the Spanish Government. A.H. received funding from the Earth System Science Research School (ESSReS), an initiative of the Helmholtz Association of German research centres (HGF) at the Alfred Wegener Institute for Polar and Marine Research. All the numerical simulations were performed with the MareNostrum Supercomputer hosted by the Barcelona Supercomputing Center. We also thank Beatriz Monge-Sanz for providing the COPCAT coefficients.



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





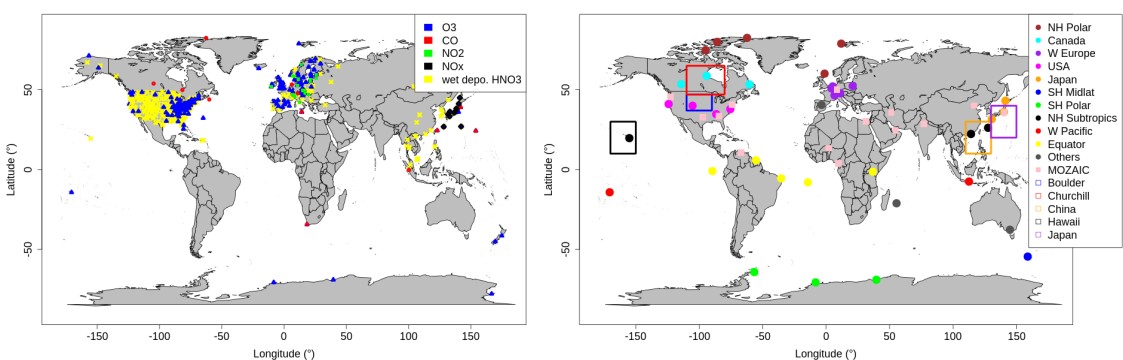

**Figure 1.** Stations used for the evaluation of the NMMB/BSC-CTM model. On the left, surface-monitoring rural stations of $O_3$ (blue triangle), CO (red circle), $NO_2$ (green square cross) and $NO_X$ (black diamond) are shown. Moreover, wet deposition $HNO_3$ (yellow cross) measurement locations are presented. On the right, locations of the different ozonesondes used ($O_3$ vertical profiles) are shown. Ozonesonde are grouped by the following regions: NH Polar (brown circle), Canada (cyan circle), W. Europe (purple circle), USA (pink circle), Japan (orange circle), SH Midlat (blue circle), SH Polar (green circle), NH Subtropics (black circle), W. Pacific (red circle), Equator (yellow circle) and Others (grey circle). In addition, CO vertical profiles from the aircraft campaign MOZAIC (pink square) are presented. Finally, large rectangles show areas for the climatology analysis ($NO_x$, PAN and $HNO_3$) of Boulder (blue), Churchill (red), China (orange), Hawaii (black) and Japan (purple).





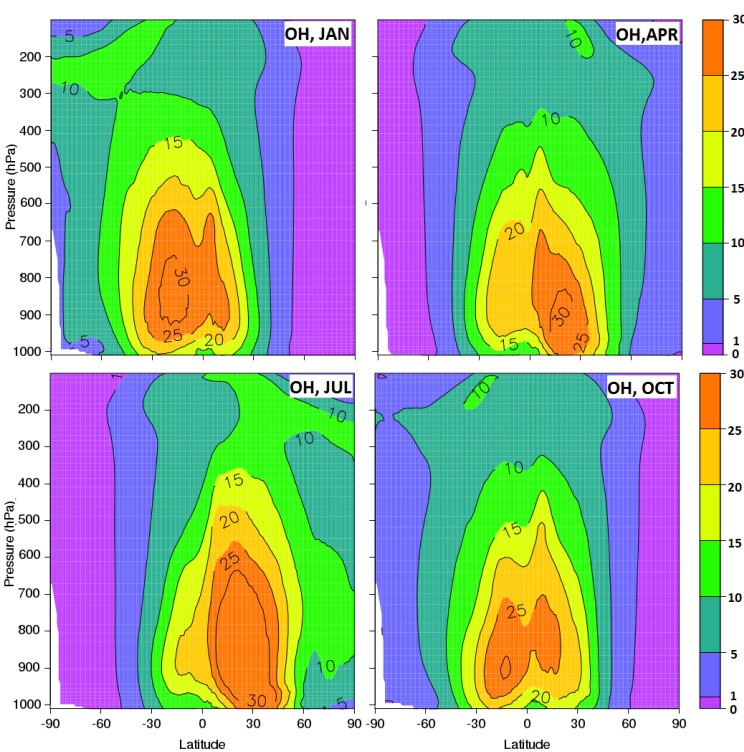

**Figure 2.** Zonally monthly mean OH concentrations ($10^5$ molecules / $cm^{-3}$) for January, April, July and October by the NMMB/BSC-CTM model





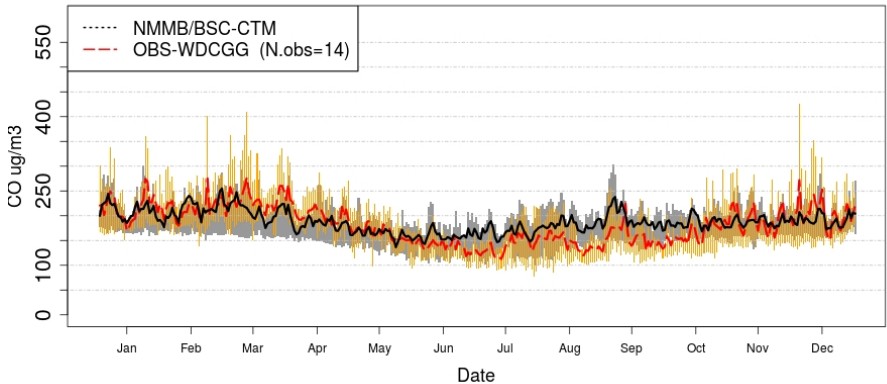

**Figure 3.** Time series of CO daily mean concentration in $\mu g\ m^{-3}$, averaged over all the rural WDCGG stations used. Observations are in a solid red line and model data in a solid black line. Bars show the 25th-75th quartile interval for observations (orange bars) and for model simulation (grey bars).

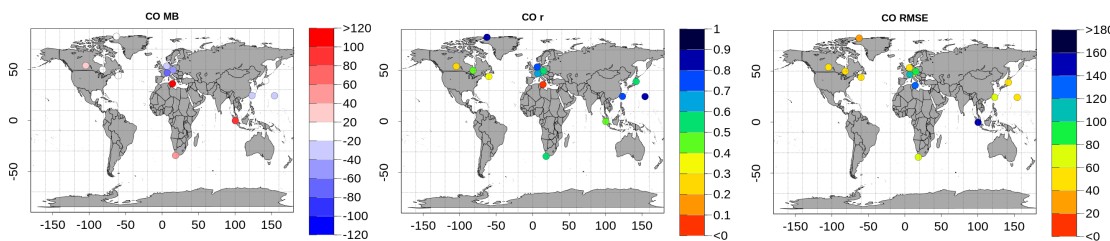

**Figure 4.** CO spatial distribution of mean bias (MB, %) (left panel) , correlation (r) (middle panel) and root mean square error (RMSE,$\mu g$ $m^{-3}$) (right panel) at all rural WDCGG stations used.




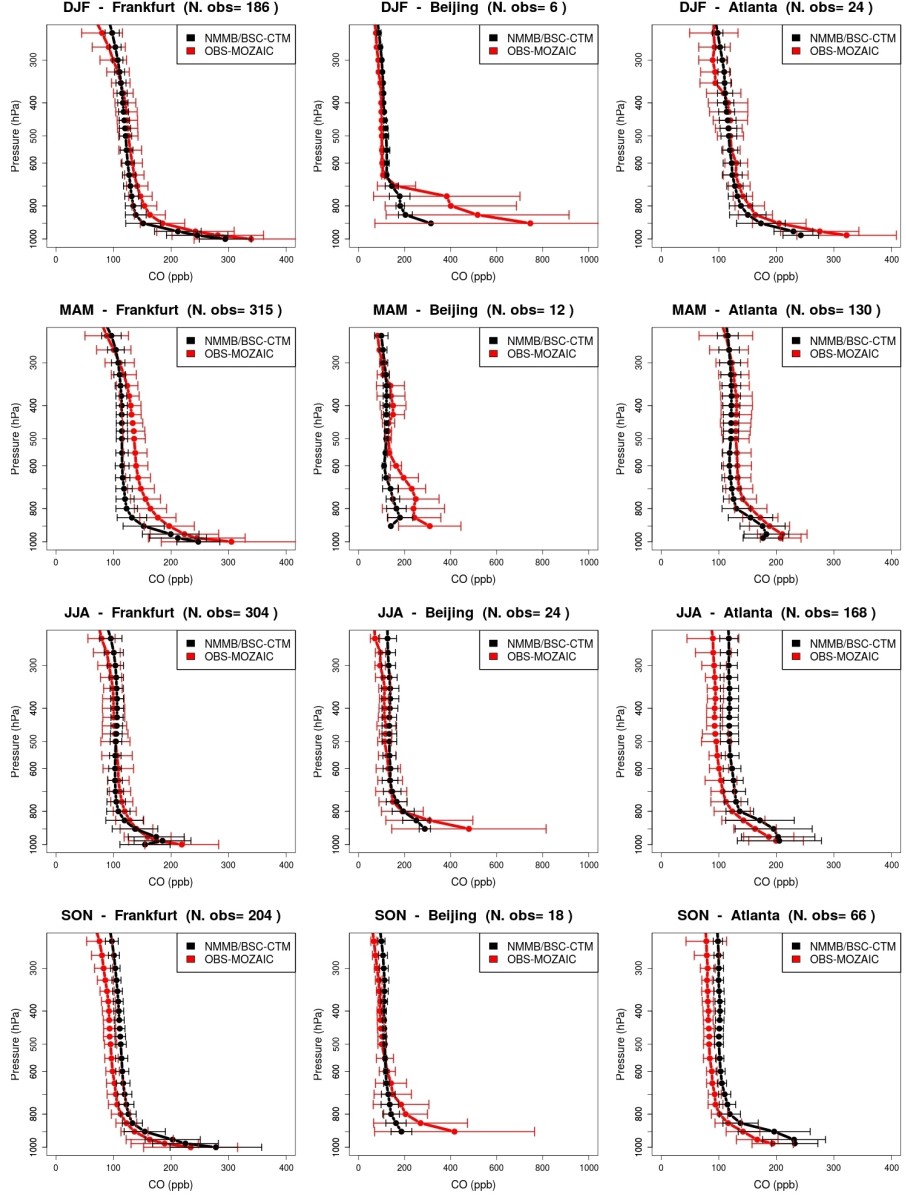

**Figure 5.** CO vertical profile seasonal averages over Frankfurt, Beijing and Atlanta (from left to right) for the whole year 2004. Observations are in a solid red line and model data in a solid black line. The number of observations flights is given on the top of each plot.




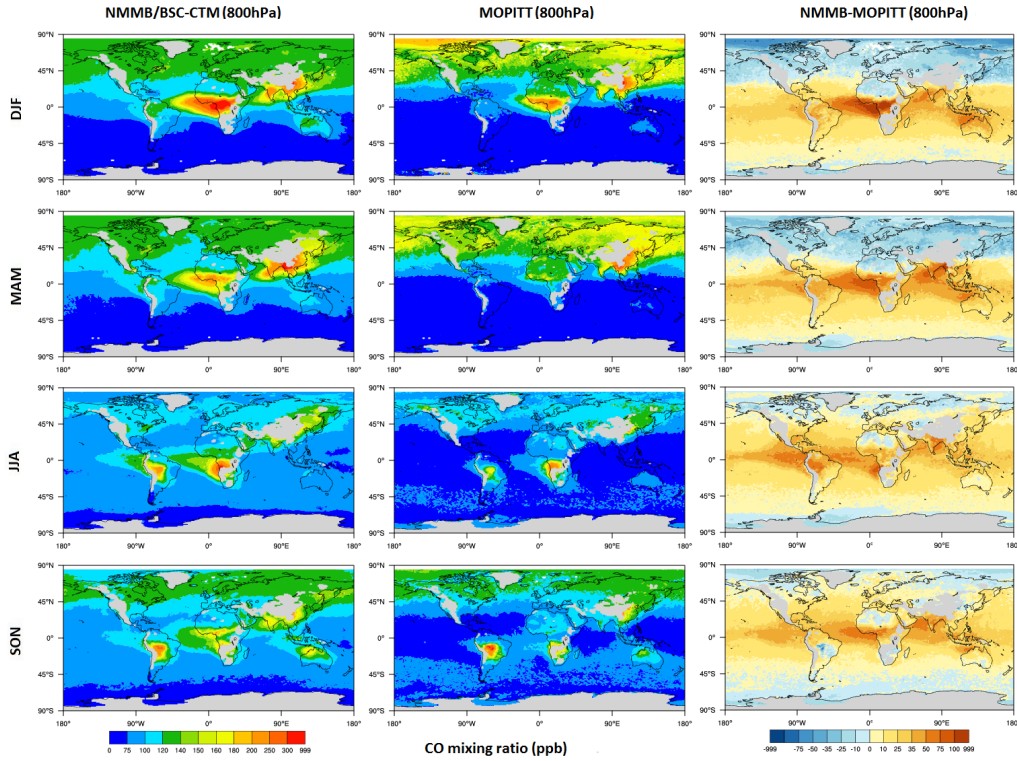

**Figure 6.** Comparison of modelled NMMB/BSC-CTM CO mixing ratio at 800 hPa against satellite data (MOPITT) for (from top) (DJF for December-January-February, MAM for March-April-May, JJA for June-July-August and SON for September-October-November) for the whole year 2004 in ppb. NMMB/BSC-CTM data is displayed in the left panel, MOPITT data in the middle panel and the bias in the right panel.





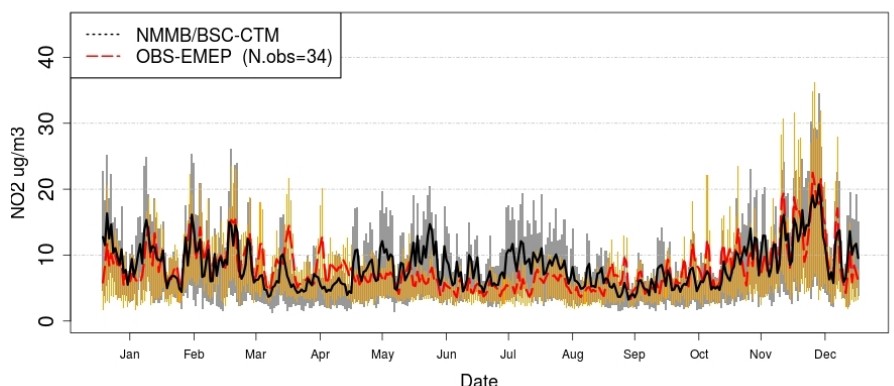

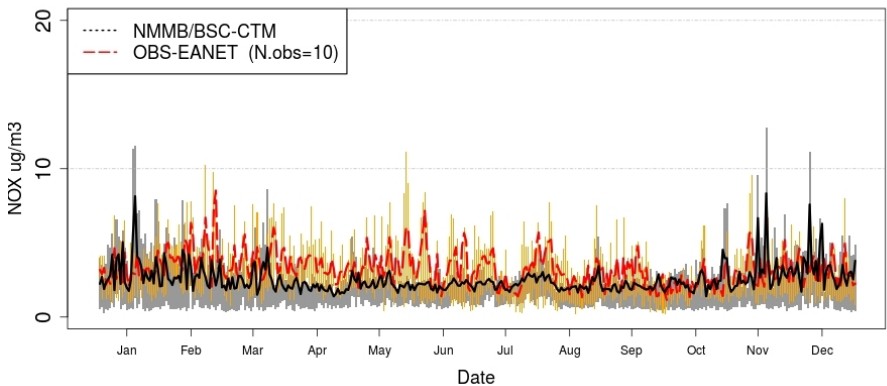

**Figure 7.** Time series of NO$_2$ (top) and NO$_x$ (bottom) daily mean concentration averaged over all the rural EMEP and EANET stations, respectively, used in $\mu$g $m^{-3}$. Observations are in a solid red line and model data in a solid black line. Bars show the 25th-75th quartile interval for observations (orange bars) and for model simulation (grey bars).





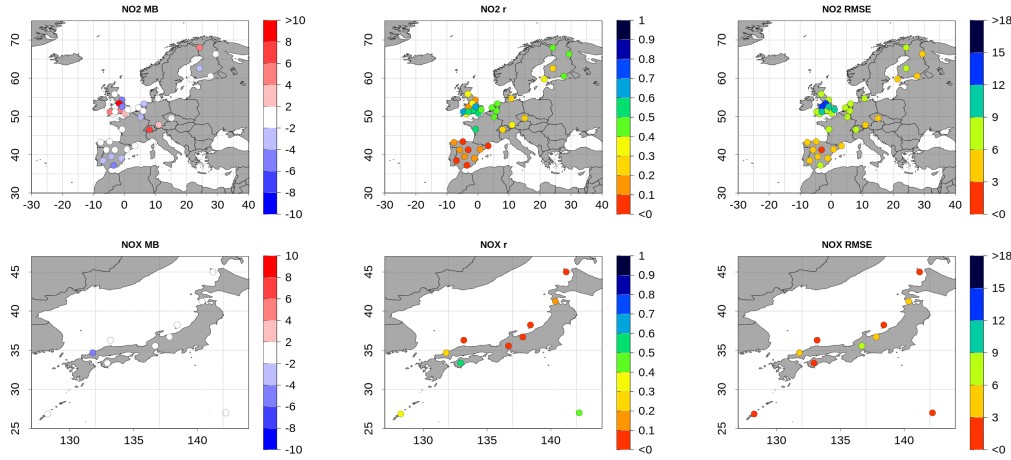

**Figure 8.** NO$_2$ (top) NO$_x$ (bottom) and spatial distribution of mean bias (MB, %) (left panel) , correlation (r) (middle panel) and root mean square error (RMSE, $\mu$g $m^{-3}$) (right panel) at all rural EMEP and EANET, respectively, stations used





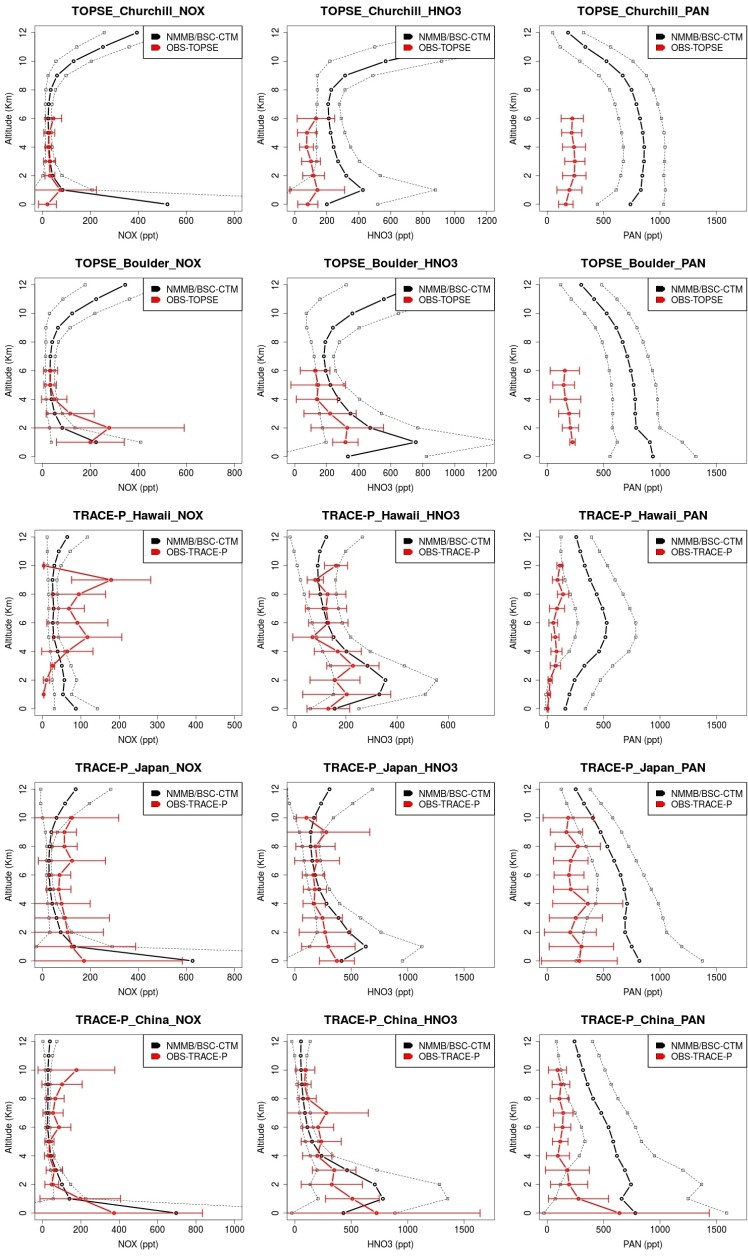

**Figure 9.** Comparison of modeled (black lines) and observed (red lines) vertical profiles of $NO_X$ (first column), $HNO_3$ (second column) and PAN (third column) for several regions over US, China, Hawaii and Japan.





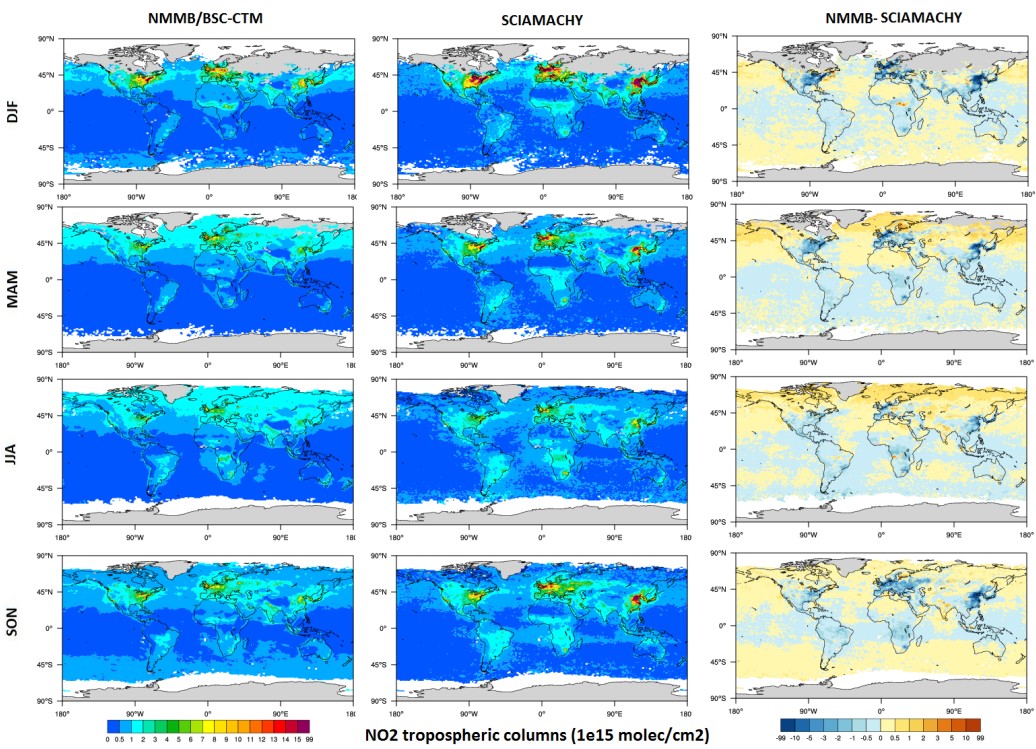

**Figure 10.** Comparison of modelled NMMB/BSC-CTM NO$_2$ vertical tropospheric columns against satellite data (SCIAMACHY) for (from top) DJF, MAM, JJA, and SON for the entire year 2004 in $1e^{15}$ molec/cm$^2$ . NMMB/BSC-CTM data is displayed in the left panel, SCIA-MACHY data in the middle panel and the bias in the right panel.




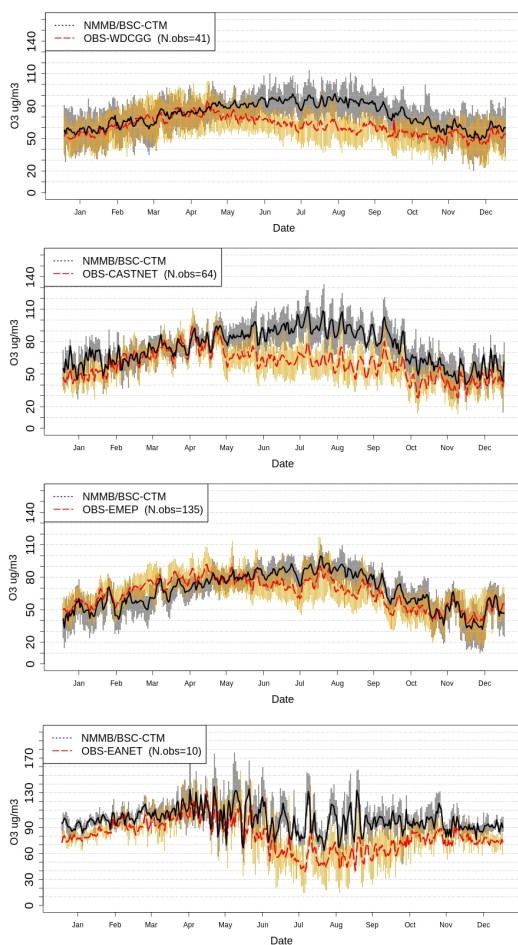

**Figure 11.** Time series of O$_3$ daily mean concentration averaged over all the rural WDCGG, CASTNET, EMEP and EANET stations (from top to bottom) used in $\mu$g $m^{-3}$. Observations are in a solid red line and model data in a solid black line. Bars show the 25th-75th quartile interval for observations (orange bars) and for model simulation (grey bars).



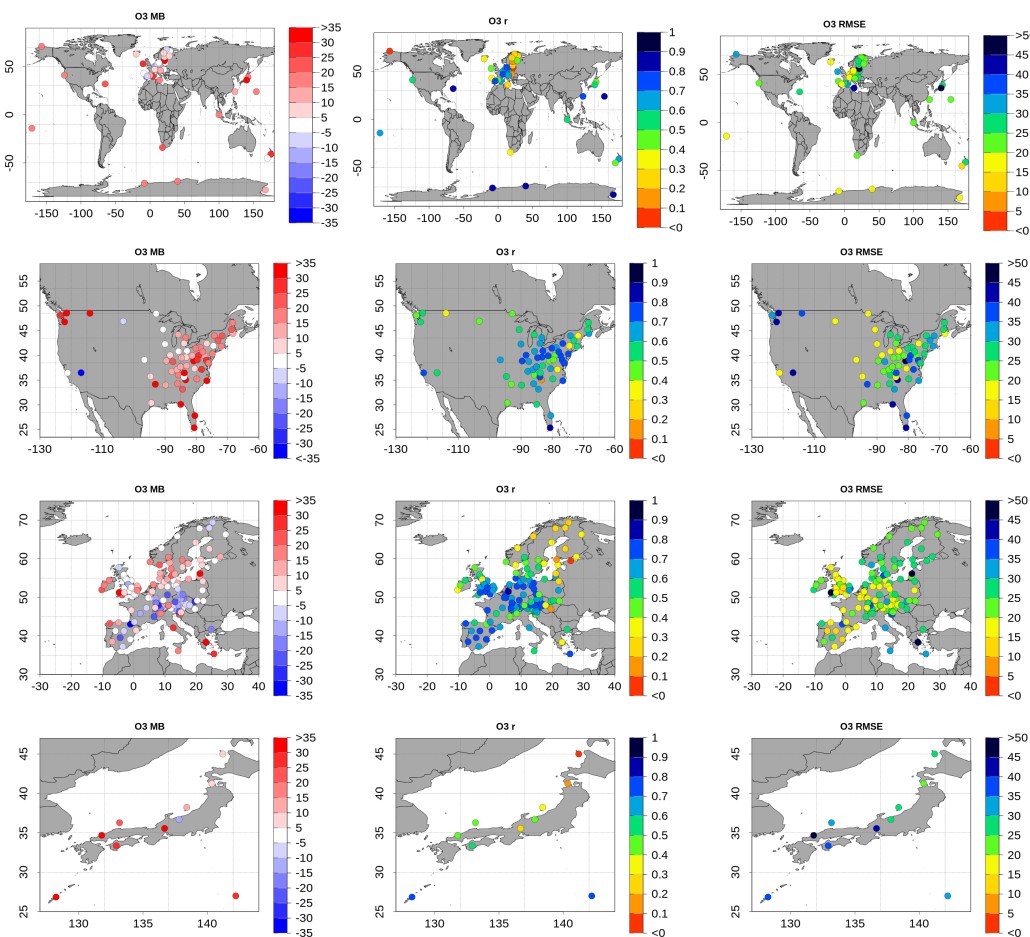

**Figure 12.** O$_3$ spatial distribution of mean bias (MB, %) (left panel) , correlation (r) (middle panel) and root mean square error (RMSE, $\mu$g $m^{-3}$) (right panel) at all rural WDCGG, CASTNET, EMEP and EANET (from top to bottom) stations used.





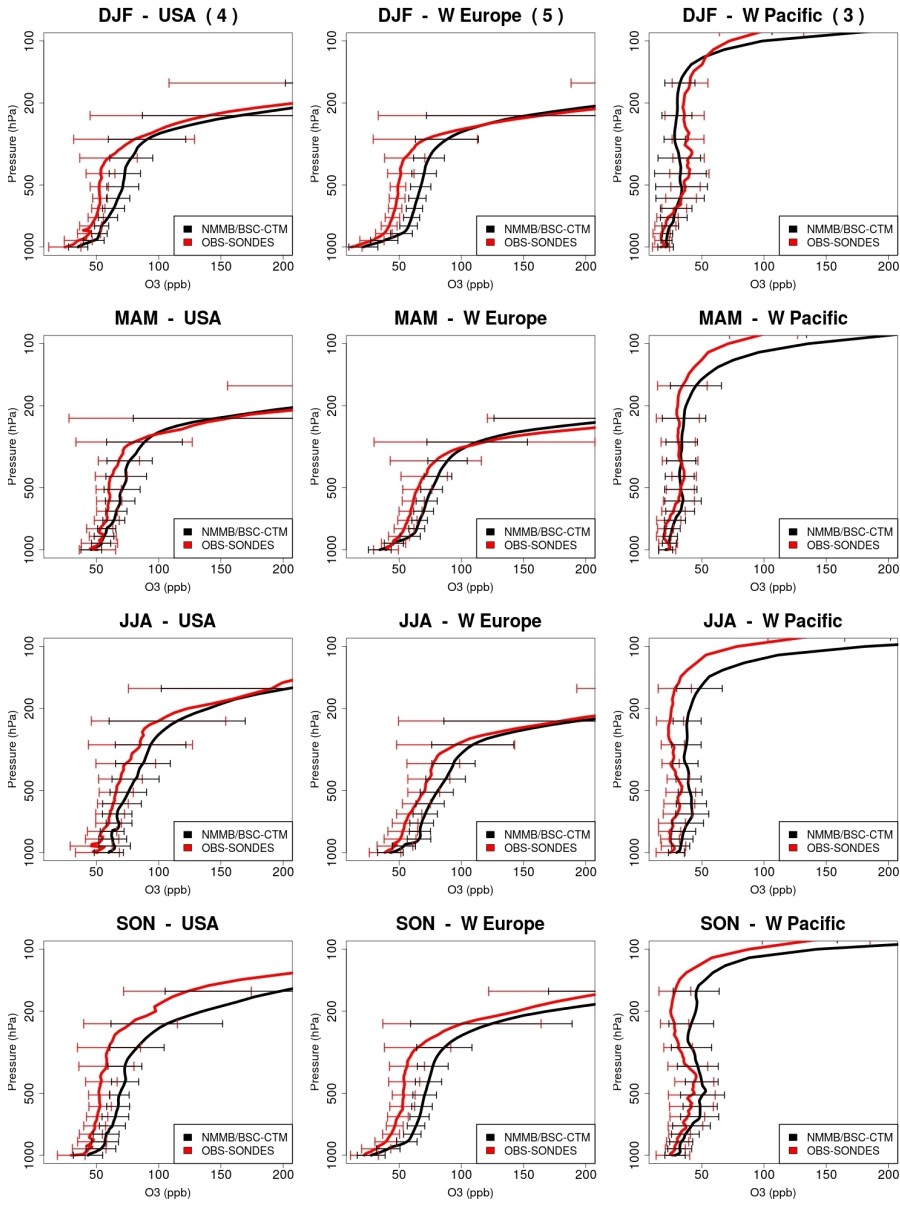

**Figure 13.** Comparison of ozonesonde measurements (red lines) and simulated (black lines) seasonal vertical profiles of $O_3$ (ppb) and standard deviations (horizontal lines). The region name and the number of stations, using brackets, are given above each plot.





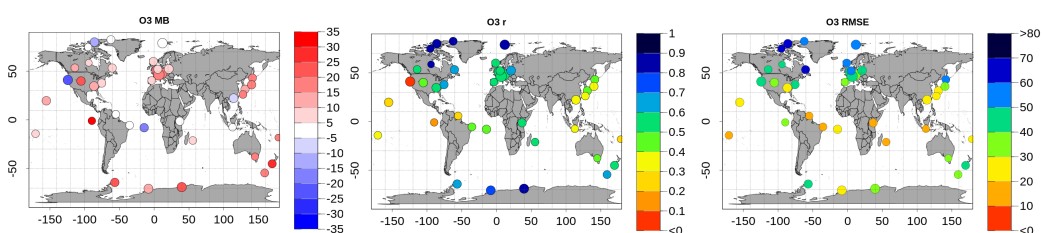

**Figure 14.** Mean tropospheric ozone bias spatial distribution of NMMB/BSC-CTM minus ozonesondes (MB, %) (left panel), root mean square error (RMSE, $\mu g\ m^{-3}$) (middle panel) and correlation (right panel) for the whole 2004, averaged between 400-1000 hPa. The diameter of the circles indicates the number of profiles over the respective stations.



**Table 1.** Model characteristics and experiment configuration

| | |
|---|---|
| **Meteorology** | |
| Dynamics | non-hydrostatic NMMB (Janjic and Gall, 2012) |
| Physics | Ferrier microphysics (Ferrier et al., 2002) |
| | BMJ cumulus scheme (Betts and Miller, 1986) |
| | MYJ PBL scheme (Janjic et al., 2001) |
| | LISS land surface model (Vukovic and Z., 2010) |
| | RRTMG radiation (Mlawer et al., 1997) |
| **Chemistry** | |
| Chemical mechanism | Carbond Bond 05 (Yarwood et al., 2005) |
| Photolysis scheme | online Fast-J photolysis scheme (Wild et al., 2000) |
| Aerosols | No aerosols considered in this study |
| Dry deposition | Wesley resistance approach from Wesely (1989) |
| Wet deposition | Grid and sub-grid scale from Foley et al. (2010) |
| Biogenic emissions | MEGAN (Guenther et al., 2006) |
| Anthropogenic and other natural emissions | ACCMIP (Lamarque et al., 2010) and POET (Granier et al., 2005) |
| Stratospheric ozone | COPCAT (Monge-Sanz et al., 2011) |
| **Resolution and Initial conditions** | |
| Horizontal resolution | $1.4^\circ \times 1^\circ$ |
| Vertical layers | 64 |
| Top of the atmosphere | 1 hPa |
| Chemical initial condition | MOZART4 (Emmons et al., 2010) |
| Meteorological initial condition | FNL/NCEP |
| Chemistry spin-up | 1 year |

**Table 2.** Emissions totals by category for 2004 in Tg(species)/year. Anthropogenic and biomass burning applied in this study are based on Lamarque et al. (2013). Ocean and soil natural emissions are based on the POET (Granier et al., 2005) global inventory. Biogenic emissions are computed online from the MEGAN (Guenther et al., 2006).

| Species | Anthrop. | Bio. burning | Biogenic | Soil | Ocean |
|---|---|---|---|---|---|
| CO | 610.5 | 459.6 | 148.13 | - | 19.85 |
| NO | 85.8 | 5.4 | 16.54 | 11.7 | - |
| $SO_2$ | 92.96 | 3.84 | - | - | - |
| Isoprene ($C_5H_8$) | - | 0.15 | 683.16 | - | - |
| Terpene ($C_{10}H_6$) | - | 0.03 | 120.85 | - | - |
| Xylenes ($C_8H_{10}$) | 1.05 | 0.16 | 1.36 | - | - |
| Methanol ($CH_3OH$) | - | - | 159.91 | - | - |
| Ethanol ($C_2H_6O$) | 4.28 | 3.7 | 17.06 | - | - |
| Formaldehyde (HCHO) | 4.24 | 0.35 | 9.58 | - | - |
| Aldehyde (R-CHO) | - | - | 5.06 | - | - |
| Toluene ($C_7H_8$) | 0.66 | 0.19 | 0.79 | - | - |
| Ethane ($C_2H_6$) | 1.27 | 0.57 | 0.48 | - | - |
| Ethylene ($C_2H_4$) | 3.32 | 2.71 | 32.03 | - | - |



**Table 3.** Ozonesondes main information used in this model evaluation for the year 2004. Location of these ozonesondes is displayed in the third and fourth columns. Columns 6-9 display the number of available measurements for each season (DJF for December-January-February, MAM for March-April-May, JJA for June-July-August and SON for September-October-November).

| Station | Country | Latitude | Longitude | Region | DJF | MAM | JJA | SON |
|---|---|---|---|---|---|---|---|---|
| Kagoshima | Japan | 31.6N | 130.6E | Japan | 13 | 12 | 11 | 12 |
| Saporo | Japan | 43.1N | 141.3E | Japan | 12 | 10 | 12 | 10 |
| Tsukubay | Japan | 36.1N | 140.1E | Japan | 14 | 13 | 12 | 12 |
| Alert | Canada | 82.5N | 62.3W | NH Polar | 11 | 10 | 13 | 9 |
| Edmonton | Canada | 53.5N | 114.1W | Canada | 7 | 12 | 10 | 10 |
| Resolute | Canada | 74.8N | 95.0W | NH Polar | 9 | 10 | 8 | 6 |
| Macquarie Island | Australia | 54.5S | 158.9E | SH Midlat | 6 | 15 | 12 | 9 |
| Lerwick | Great Britain | 60.1N | 1.2W | W Europe | 9 | 13 | 13 | 12 |
| Uccle | Belgium | 50.8N | 4.3E | W Europe | 35 | 37 | 36 | 36 |
| Goose Bay | Canada | 53.3N | 60.4W | Canada | 12 | 13 | 12 | 12 |
| Churchill | Canada | 58.7N | 94.1W | Canada | 7 | 6 | 4 | 8 |
| NyAlesund | Norway | 78.9N | 11.9E | NH Polar | 25 | 24 | 23 | 17 |
| Hohenpeissenberg | Deutschland | 47.8N | 11.0E | Europe | 34 | 34 | 26 | 31 |
| Syowa | Japan (Antarctica) | 69.0S | 39.6E | SH Polar | 16 | 16 | 19 | 26 |
| Wallops Island | USA | 37.9N | 75.5W | USA | 11 | 15 | 17 | 7 |
| Hilo | USA | 19.7N | 155.1W | NH Subtropic | 13 | 18 | 14 | 12 |
| Payerne | Switzerland | 46.5N | 6.6E | Europe | 38 | 40 | 38 | 40 |
| Nairobi | Kenya | 1.3S | 36.8E | Equador | 11 | 13 | 13 | 13 |
| Naha | Japan | 26.17N | 127.7E | NH Subtropics | 9 | 12 | 8 | 10 |
| Samoa | Independent State of Samoa | 14.2S | 170.6W | W Pacific | 9 | 11 | 8 | 9 |
| Legionowo | Poland | 52.4N | 20.9E | Europe | 16 | 18 | 16 | 18 |
| Marambio | Antarctica | 64.2S | 56.6W | SH Polar | 10 | 7 | 15 | 22 |
| Lauder | New Zealand | 45.0S | 169.7E | SH Midlat | 11 | 13 | 13 | 9 |
| Madrid | Spain | 40.5N | 3.6W | Others | 11 | 9 | 8 | 12 |
| Eureka | Canada | 80.0N | 85.9W | NH Polar | 17 | 17 | 11 | 13 |
| De Bilt | Nederland | 52.1N | 5.2E | Europe | 13 | 10 | 14 | 12 |
| Neumayer | Antarctica | 70.7S | 8.3W | SH Polar | 11 | 13 | 13 | 31 |
| Hong Kong | China | 22.3N | 114.2E | NH Subtropics | 12 | 26 | 11 | 13 |
| Broad Meadows | Australia | 37.7S | 144.9E | Others | 6 | 7 | 7 | 11 |
| Huntsville | USA | 34.7N | 86.6W | USA | 14 | 13 | 23 | 13 |
| Parambio | Surinam | 5.8N | 55.2W | Equador | 11 | 8 | 9 | 9 |
| Reunion Island | France | 21.1S | 55.5E | Others | 9 | 14 | 9 | 6 |
| Watukosek | Indonesia | 7.5S | 112.6E | W Pacific | 7 | 11 | 10 | 6 |
| Natal | Brasil | 5.5S | 35.41W | Equador | 10 | 12 | 13 | 7 |
| Ascencion Island | Great Britain | 7.98S | 14.42W | Equador | 12 | 12 | 12 | 18 |
| San Cristobal | Galapagos | 0.92S | 89.6W | Equador | 7 | 4 | 10 | 13 |
| Boulder | USA | 40.0N | 105.26W | USA | 12 | 11 | 17 | 16 |
| Trinidad Head | USA | 40.8N | 124.2W | USA | 4 | 7 | 5 | 8 |
| Suva | Fiji | 18.13S | 178.4E | W Pacific | 13 | 12 | 48 | 11 |





**Table 4.** MOZAIC aircraft information used in this model evaluation for the year 2004. Location of the MOZAIC measurements is displayed in the third and fourth columns. Columns 5-8 display the number of available measurements for each season (DJF for December-January-February, MAM for March-April-May, JJA for June-July-August and SON for September-October-November).

| Station | Country | Latitude | Longitude | DJF | MAM | JJA | SON |
|---|---|---|---|---|---|---|---|
| Abu Dhabi | United Arab Emirates | 24.44N | 54.65E | 11 | 17 | 58 | 20 |
| Atlanta | USA | 33.63N | 84.44W | 24 | 130 | 168 | 66 |
| Beijing | China | 40.09N | 116.6E | 5 | 12 | 23 | 17 |
| Cairo | Egypt | 30.11N | 31.41E | 19 | 16 | 2 | 8 |
| Caracas | Venezuela | 10.6N | 67W | 21 | 9 | 9 | 21 |
| Dallas | USA | 32.9N | 97.03W | 8 | 24 | 24 | 10 |
| Douala | Cameroon | 4.01N | 9.72E | 7 | 0 | 10 | 6 |
| Frankfurt | Germany | 50.02N | 8.53E | 169 | 295 | 286 | 192 |
| New Delhi | India | 28.56N | 77.1E | 30 | 24 | 72 | 38 |
| New York | USA | 40.7N | 74.16W | 79 | 23 | 41 | 16 |
| Niamey | Niger | 13.48N | 2.18E | 4 | 0 | 12 | 12 |
| Portland | USA | 45.59N | 122.6W | 5 | 8 | 5 | 4 |
| Tehran | Iran | 35.69N | 51.32E | 8 | 11 | 31 | 18 |
| Tokyo | Japan | 35.76N | 140.38E | 38 | 50 | 56 | 34 |

**Table 5.** Description of additional aircraft campaign data. Location of the measurements campaigns is displayed in the third and fourth columns. The fifth column lists the date of these campaigns.

| Region Name | Expedition | Latitude | Longitude | Date |
|---|---|---|---|---|
| Boulder | TOPSE | 37-47N | 110-90W | 5 February to 23 May 2000 |
| Churchill | TOPSE | 47-65 N | 110-80W | 5 February to 23 May 2000 |
| China | TRACE-P | 10-30N | 110-130E | 24 February to 10 April 2001 |
| Hawaii | TRACE-P | 10-30N | 170-150W | 24 February to 10 April 2001 |
| Japan | TRACE-P | 20-40N | 130-150E | 24 February to 10 April 2001 |

**Table 6.** Annual mean burden of tropospheric CO (Tg CO) in NMMB/BSC-CTM, MOZART-2 and TM5 global models

| Model | Burden | | | | | | Dry depo. | Reference |
|---|---|---|---|---|---|---|---|---|
| | Global | NH | SH | Trop. | N. Extratrop. | S. Extratrop. | | |
| NMMB/BSC-CTM | 399 | 221 | 177 | 229 | 101 | 67 | 24 | This study |
| MOZART-2 | 351 | 210 | 142 | 199 | 102 | 50 | 2 | Horowitz et al. (2003) |
| TM5 | 353 | - | - | 188 | 106 | 59 | 184 | Huijnen et al. (2010) |
| C-IFS | 361 | - | - | - | - | - | - | Flemming et al. (2015) |

**Table 7.** Annual mean burden, dry deposition of tropospheric O$_3$ and stratospheric inflow (Tg O$_3$) for the NMMB/BSC-CTM, MOZART-2, TM5 and LMDz-INCA, C-IFS global models, and two different Multimodel ensembles (25 and 15 global models).

| Model | Burden | | | | | | Dry deposition | Stratospheric inflow | Reference |
|---|---|---|---|---|---|---|---|---|---|
| | Global | NH | SH | Trop. | N. Extra. | S. Extra. | | | |
| NMMB/BSC-CTM | 348 | 189 | 158 | 171 | 101 | 75 | 1201 | 384 | This study |
| MOZART-2 | 362 | 203 | 159 | 203 | 99 | 60 | 857 | 343 | Horowitz et al. (2003) |
| TM5 | 312 | - | - | 165 | 84 | 63 | 829 | 421 | Huijnen et al. (2010) |
| LMDz-INCA | 303 | 178 | 125 | - | - | - | 1261 | 715 | Folberth et al. (2006) |
| C-IFS | 390 | - | - | - | - | - | - | - | Flemming et al. (2015) |
| Multimodel | 344 ± 39 | - | - | - | - | - | 1003 ± 200 | 552 ± 168 | Stevenson et al. (2006) |
| Multimodel | 337 ± 23 | - | - | - | - | - | - | - | Young et al. (2013) |