# Peer review of "Gas-phase chemistry in the online multiscale NMMB/BSC Chemical Transport Model: Description and evaluation at global scale"

_Geoscientific Model Development, 2016_

## Short Comment (SC1) · 16 Jun 2016

Dear authors,

In my role as Executive editor of GMD, I would like to bring to your attention our Editorial version 1.1:

http://www.geosci-model-dev.net/8/3487/2015/gmd-8-3487-2015.html

This highlights some requirements of papers published in GMD, which is also available on the GMD website in the 'Manuscript Types' section:

http://www.geoscientific-model-development.net/submission/manuscript_types.html

[Figure]

In particular, please note that for your paper, the following requirement has not been met in the Discussions paper:

- "The main paper must give the model name and version number (or other unique identifier) in the title."

Please add the version number for your model to which the description and evaluation applies in the title upon your revised submission to GMD.

Yours,

Astrid Kerkweg

---

## Referee Comment (RC1) · Anonymous Referee #1 · 1 Aug 2016

The authors present a detailed description and evaluation of the tropospheric chemistry transport model NMMB/BSC-CTM. This model domain has been expanded from regional to global. The focus of their evaluation is gas-phase chemistry with emphasis on tropospheric ozone and its precursors. Several ground-based, aircraft and satellite data are used to show model strengths and weaknesses. The paper is well-written and is within the scope of the journal. I would recommend the publication of this paper after my minor comments below have been addressed:

Page 2, line 15: replace "fed by emission inventories" with "emissions of chemical species".

Page 2, Line 31: Define NMMB/BSC here.

[Figure]

Page 3, line 19: replace "main reactions occurring in the atmosphere by" with "atmospheric composition". It would be helpful to give a motivation for choosing year 2004 for evaluation.

Page 3, line 28: Insert "direct" before radiative effect.

Page 5, line 10: A reference is needed here.

Page 5, line 17: 1.85 ppm is too high for this year of simulation. Can you please provide a justification for using this number? Also, is the CH4 concentration constant throughout the troposphere in the model or only at the surface?

Page 8, section 2.2.6: Where does the MEGAN model implemented in this CTM derive the leaf-area index needed to calculate biogenic emissions?

Page 8, line 29: Need a period after "parameters".

Page 9, section 3: What is the size of the bottom-most layer in the model? Also provide an estimate of the time it takes to run a year's simulation.

Page 9, lines 12-13: Since emissions after year 2000 were not prvided by Lamarque et al., which projection (RCP?) was used for 2010 emissions to perform linear interpolation?

Page 10, line 25: Any particular reason why only two aircraft campaigns were used for the evaluation instead of several others available in Emmons et al. (2000).

Page 12, section 5.1: What is the simulated tropospheric lifetime of methane in the model and how does it compare with that from multi-model studies (e.g., Naik et al 2013a)? How does the simulated OH interhemispheric ratio compare with other studies (Naik et al., 2013a; Patra et al., 2014). Lightning NOx emissions have been shown to contribute significantly to tropospheric OH concentrations (Murray et al., 2013). Lightning NOx emissions are not considered in the current model set-up. Please explain how the simulated OH concentrations match closely with those of other modeling studies that include lightning NOx emissions.

Page 12, line 28: Need a reference for aerosol influence on OH. Also, a larger oxidizing capacity would be simulated if lightning NOx emissions were included in these simulations.

Page 13, line 9-10: Please give the reason why there were low CO concentrations in 2004 despite large Alaskan and Canadian wildfires.

Page 13, 11-15: Other modeling studies suggest even lower CO burden (e.g., Naik et al., 2013b). Could higher CH4 concentration prescribed in the model play a role in the simulated high CO burden?

Page 13, last paragraph: the role of seasonal CO emissions in explaining the low northern hemisphere wintertime bias has been highlighted by Stein et al., (2014), which should be noted here.

Page 15, line 25: Give the lifetime of NOx.

Page 18, line 13-18: How do the calculated dry deposition estimates compare with those from more recent chemistry-climate model simulations (e.g., Naik et al., 2013b).

Page 19: What fraction of model O3 biases could be related to biases in the simulated meteorological fields (e.g., temperature)?

Map figures: Please remove the grey background on the maps as this makes it difficult to read the colours.

Figure 6, 10: Colour bar text is too small to read.

References: Murray LT, et al. (2013) Interannual variability in tropical tropospheric ozone and OH: The role of lightning. J Geophys Res Atmos 118(19):11,468–11,480, doi:10.1002/jgrd.50857. Naik et al., (2013a), Preindustrial to present day changes in tropospheric hydroxyl radical and methane lifetime from the Atmospheric Chemistry and Climate Model Intercomparison Project (ACCMIP), Atmos. Chem. Phys., 13,

5277-5298, doi:10.5194/acp-13-5277-2013. Naik et al., 2013b, Impact of preindustrial to present day changes in short-lived pollutant emissions on atmospheric composition and climate forcing, J. Geophys. Res., doi: 10.1002/jgrd.50608. Patra et al., (2014), Observational evidence interhemispheric hydroxyl-radical parity, Nature, doi:10.1038/nature13721. Stein et al., (2014) On the wintertime low bias of Northern Hemisphere carbon monoxide found in global model simulations, Atmos. Chem. Phys., 14, 9295-9316, doi:10.5194/acp-14-9295-2014.

---

## Referee Comment (RC2) · Anonymous Referee #2 · 2 Aug 2016

Review of

"Gas-phase chemistry in the online multiscale NMMB/BSC Chemical Transport Model: Description and evaluation at global scale" by Badia et al.

Overview:

The paper is a description and an evaluation of a one-year (2004) global simulation of what is called the NMMB/BSC Chemical Transport Model. The model results for CO, O3 and NO2/PAN/HNO3 are comparted against surface observations, profile observations and satellite retrievals. The comparison shows that the NMMB/BSC model gives acceptable results but both CO and ozone are overestimated on the global scale.

[Figure]

General remarks:

My two main concerns with the presented model run for 2004 is (i) that NOx emissions from lightning were not considered in the model run and (ii) that no specific biomass burning emissions for 2004 were used. A parameterisation of lightning emissions is scientific standard in global CTMs and there is no good reason, why such an important contribution to global tropospheric chemistry can be omitted. Getting daily or 8-day-mean 2004 biomass emission data (GFED, GFAS etc.) would not have been difficult. Also the lack of seasonality of the anthropogenic emissions is an unnecessary simplification. Against the backdrop of these omissions it becomes difficult to draw conclusion from the model results and it severely undermines the scientific credibility of the paper.

On the other hand, the presented model had two advanced properties, namely the on-line calculation of VOC emissions using the MEGAN model and the fact that the presented model is an on-line coupled chemistry – meteorological model (or Chemistry-GCM). The term CTM, which the authors choose, is commonly used for off-line model without the simulation of meteorology (see Baklanov et al. 2014, ACP). I therefore recommend not to use the term CTM in the name of the model because it is an on-line coupled model. Unfortunately, it is a missed chance that these two new aspects were not explored further in the paper.

The evaluation is carried out with a well-balanced choice of observations but the results are too often only described with the words such as "good agreement" etc. I think this is not very meaningful, instead the results should be quantified in a better way, i.e. a bias of 10 ppb, 20% etc.

It is a thought-provoking result that both CO and ozone are overestimated because an overestimation of the oxidation capacity is often linked with CO underestimation (see Strode et al. 2015, ACP) It is something which can not be found in other models using similar emission data, and especially for models that also use the CB05 chemical mechanism. I think this result deserves a more thorough investigation.

[Figure]

Without carrying out sensitivity studies, it is in general problematic to come to valid conclusion on the reasons for certain aspects (bad or good) of the model performance. The authors predominately only argue (without doing sensitivity studies) that (i) deficiency in the emissions and/or (ii) the lack of considering aerosol in the photolysis rates are the reasons for identified model deficiencies. While there is consensus in the scientific community that emissions can be very uncertain, there is no evidence given in the paper, why the aerosol impact should be so important as the authors claim. (I am happy to be convinced otherwise by a sensitivity study or a reference to it).

The authors should discuss other aspect of their model setup in more detail. If there is the feeling that photolysis rates play a role, then cloud cover would be the first suspect. The cloud cover should be checked for biases since the model simulates clouds itself. Also worth checking are the ozone total columns used in the photolysis scheme because they are not constrained by observations.

Another potentially important aspect is the fact the emissions are injected uniformly in the lowest 500m (anthropogenic) or 1300 m (biomass burning). This could have a large impact on dry deposition, which depends on the surface level concentration, and ozone titration by NO during the night. The 500 m seems to be an exaggeration of the extent of the mixed layer during the night over land and the choice needs to be better motivated. One would expect that the diffusion scheme of the model simulates the vertical mixing in the PBL. Also, the 1300 m for the biomass burning injection would need to be justified, as the fire injection height can vary substantially (see for example. Remy al., 2016, ACP)

The paper would greatly benefit from proofreading for English language.

Specific comments:

P1

Title: consider not calling the model a CTM as CTM's are understood as "off-line"

Abstract - Spell out NMMB/BSC and other acronyms - No need to specify all the network and instruments in the abstract

P2

L 27: a better reference for IFS-MOZART is Flemming et al. 2009, GMD

L 33: Please clarify if the non-hydrostatic option was used in the run.

P3

L 26: better "in detail"

P4

L15-25: please clarify which of the options is actually used in the presented run. The other options don't need to be mentioned. They could be referenced.

P 5

L 17: 1850 ppb of methane seems too high for 2004. The value should be 1775 ppb http://www.esrl.noaa.gov/gmd/ccgg/trends_ch4/

L 22: Add more information about the realism of the two input fields (overhead ozone and clouds)

P 6

L 10: Is this a monthly climatology ?

L 13: "cloud processes" – does this also include wet-phase chemistry ?

L 14: The presented terms are not clear. Please clarify what you mean by all the mentioned processes. For example, what is wet deposition for non-precipitating cloud?

L16: Why only in-cloud scavenging and not all the other processes ?

L 19: How is the cloud time scale derived?

P7

L2: Do you refer to large-scale and convective precipitation here?

L7: "Convective mixing" do you mean transport by convective mass fluxes ?

L 13: 100 hPa is a rather high tropopause for mid- and high latitudes.

L 13: Were these Mozart 4 fields evaluated for the stratosphere ?

L 18ff: There is no need to present the COPCAT scheme here, a reference to the paper is enough.

L 18: Please provide information on the biases of your stratosphere ozone simulated by the COPCAT scheme because they have an impact on the photolysis rates.

P 8

L 14: No lightning emissions is a severe shortcoming of the simulation and the paper (see my general comment).

L 15: Please explain in more detail how the MEGAN code was integrated in your model.

L34: I don't understand the 24h averages here. I thought (L23) the actual hourly meteorological data were used for the calculation of the Megan emissions.

P 9

L3: better say "every 720 s"

L 5: Which fields are initialised (also clouds or only T, v,w,q). What is known about the biases of the 24 h forecasts?

L 12: not using 2004 fire emissions is a severe omission (see my general comment). Please clarify what fire emissions have been used. Was it an average for the period? It is not clear what "interpolated" means. Do the fire emissions have a seasonal cycle ?

L 26: see my general comment, please justify the choices

L 29: Please provide reference the strong impact of aerosol on the photolysis rates.

P 11

L27: It is not clear how missing data in the surface observations were considered. If you compare only averages without timely match give numbers of the amount of missing data.

L 30: 1000m asl.? This could be a mountain stations near to the coast or a station on a flat plateau inland. It would be better to include the model orography in the choice of the mountain stations. (say 500 m above orography)

P 12

L1: This choice of the tropopause is not consistent with the choice of the tropopause for the chemical boundary conditions (P7L13).

L 29: see my general comment. The aerosol effect may not be the most important one. There are many other possible explanations: high CH4, water vapour, clouds and photolysis, excessive mixing of emissions etc.

P 13

L1: CH4 is also a CO source. Please also reformulate the sentence.

L 14 add reference for C-IFS

L 29: Could the high methane be a reason ?

L 30: Figure 3. Please show separate plots for NH, SH mid-latitudes and tropics. The seasonality is obscured by averaging over all stations.

P 14

L2: no seasonality of the anthropogenic emissions is an oversimplification.

L3: Figure 3 shows the relative bias (%), not MB as defined in the supplement.

L3 Please clarify correlation of which time scale is shown, i.e. of the hourly, daily monthly values? Did you filter out seasonality? How important is the diurnal cycle to the correlation.

L 28 Stein et al. and many other authors find a general underestimation in winter and spring NH.

P 15

L8: What do you mean by overestimated emissions above the PBL ?

L 8: Please discuss the role of convection

L 32: What regime (rural, urban) was used ?

P 16

L 8: Please discuss also PBL mixing during the night

P 17

L 8: see my general comment on the use of "good agreement"

L 17: Pease mention the value of the biases.

L 26: What is the seasonal cycle of the biomass emissions ?

P 18

L 1 "rural" (?) perhaps better remote

L 4: dominated by the tropics - perhaps simply because they are the largest region on earth (?)

L 7: TM5 has a similar chemical mechanism. Should it not be similar ?

L 16: Please clarify how the STE is calculated in your model.
L 27: "all day long " ? Do you mean "throughout the year"

L 27: For global models the values are commonly given in volume mixing ratios (ppb). Try to avoid mg/m3 throughout the paper.

L 28: The emission injection (500m) leads to a dilution of NO and therefore a reduction of the ozone titration. This could also explain the overestimation.

P 19

L4 Please quantify biases, what do you mean by "error".

L 15. This points to biases of the COPCAT ozone, which has consequences for the photolysis rates.

P 20

L 8: The lack of aerosol modulation of photolysis is a probably a minor aspect. Lack of heterogeneous chemistry (N2O5) might be more important. Please also mention the main shortcomings of this simulation: (1) no lightning, (2) no 2004 biomass burning emissions and (3) no seasonal cycle for anthropogenic emissions.

L15: The paper provides no evidence for this claim - it can therefore not be a conclusion.

L 34 see above, no evidence in the paper

P 21

L 13 better "megacities"

Figure 3: better to have plots for different regions (NH, SH, Tropics) to better see the seasonal cycle. Choose a smaller y-range for more clarity. Show plot in ppb (as for the profiles) rather than microgramm/m3.

Figure 4: MB (see supplement) is defined without scaling (i.e. not relative in %). Please show the MB as defined in the supplement. Use ppb as unit.

Figure 7: better y-range, use ppb

Figure 8: as for Figure 4

Figure 11: use ppb

Figure 12: see Figure 4

Figure 13: choose x-range 0-100 for better clarity in the troposphere.

Figure 14: see Figure 4
* * *

---

## Author Comment (AC1) · 11 Nov 2016

Reply to SC1: 'Executive Editor Comment on "Gas-phase chemistry in the online multiscale NMMB/BSC Chemical Transport Model: Description and evaluation at global scale"', Astrid Kerkweg, 16 Jun 2016

Dear Executive editor of GMD,

Following the Editorial guidelines we have included the model version in the title.

Note that we have decided to rename our model following a comment from reviewer #2 about avoiding the use of CTM for an online model. Thus, the new name is NMMB-MONARCH, where MONARCH stands for "Multiscale Online Nonhydrostatic Atmo-
spheRe CHemistry model". In the responses to the reviewer's comments we keep the NMMB/BSC-CTM name to keep consistency with the manuscript submitted to GMDD, but in the revised manuscript the new name, NMMB-MONARCH, is used.

Now, the revised manuscript is entitled "Description and evaluation of the Multiscale Online Nonhydrostatic AtmospheRe CHemistry model (NMMB-MONARCH) version 1.0: gas-phase chemistry at global scale".

---

## Author Comment (AC2) · 11 Nov 2016

**RC1: 'Reviewer Comments', Anonymous Referee #1, 01 Aug 2016**

The authors present a detailed description and evaluation of the tropospheric chemistry transport model NMMB/BSC-CTM. This model domain has been expanded from regional to global. The focus of their evaluation is gas-phase chemistry with emphasis on tropospheric ozone and its precursors. Several ground-based, aircraft and satellite data are used to show model strengths and weaknesses. The paper is well-written and is within the scope of the journal. I would recommend the publication of this paper after my minor comments below have been addressed:

*Response: The authors wish to thank anonymous reviewer #1 for his/her valuable comments and suggestions.*

*Note that we have decided to rename our model following a comment from reviewer #2 about avoiding the use of CTM for an online model. Thus, the new name is NMMB-MONARCH, where MONARCH stands for "Multiscale Online Nonhydrostatic AtmospheRe CHemistry model". In the responses to the reviewer's comments we keep the NMMB/BSC-CTM name to keep consistency with the manuscript submitted to GMDD, but in the revised manuscript the new name, NMMB-MONARCH, is used.*

*Now, the revised manuscript is entitled "Description and evaluation of the Multiscale Online Nonhydrostatic AtmospheRe CHemistry model (NMMB-MONARCH) version 1.0: gas-phase chemistry at global scale".*

Page 2, line 15: replace "fed by emission inventories" with "emissions of chemical species".
*Response: Amended.*

Page 2, Line 31: Define NMMB/BSC here.
*Response: In the revised manuscript MONARCH is now defined there.*

Page 3, line 19: replace "main reactions occurring in the atmosphere by" with "atmospheric composition". It would be helpful to give a motivation for choosing year 2004 for evaluation.
*Response: Amended. 2004 is a reference year for our modeling group that we already considered in previous studies (e.g., Pay et al. 2010; Baldasano et al. 2011). Therefore, our choice is based on the amount and variety of quality controlled and quality assured observations available in our group. We don't think this information is relevant for the manuscript.*

*Pay, M. T., et al. "A full year evaluation of the CALIOPE-EU air quality modeling system over Europe for 2004." Atmospheric Environment 44.27 (2010): 3322-3342.*
*Baldasano, J. M., et al. "An annual assessment of air quality with the CALIOPE modeling system over Spain." Science of the Total Environment 409.11 (2011): 2163-2178.*

Page 3, line 28: Insert "direct" before radiative effect.
*Response: Amended.*

Page 5, line 10: A reference is needed here.
*Response: The reference of Yarwood (2005) is now included.*

Page 5, line 17: 1.85 ppm is too high for this year of simulation. Can you please provide a justification for using this number? Also, is the CH4 concentration constant throughout the troposphere in the model or only at the surface?

*Response: Considering a global background concentration of methane is a common practice in air quality modeling. Following this approach, current practices set the methane background level either as a default background concentration (i.e. 1.76 ppm, e.g. Shindell et al, 2006), or as the background level for the Northern Hemisphere (i.e. 1.85 ppm, used for instance within CMAQ). Including either of those concentrations would lead to differences with respect to reality, and we decided to select the latter, which on the other hand is the closest to the present time global background concentration (1.83 ppm, see WMO 2015, or Dlugokencky, 2016). The global average for 2004 is reported to be 0.06 to 0.07 ppm lower, around 4%. This small difference is not expected to cause any sizeable differences in the results shown here.*

*Shindell et al. Multimodel simulations of carbon monoxide: Comparison with observations and projected near-future changes. Journal of Geophysical Research Atmospheres. VOL. 111, D19306, doi:10.1029/2006JD007100, 2006*
*WMO Greenhouse Gas Bulletin Nº 11: November 2015*
*Ed Dlugokencky, NOAA/ESRL www.esrl.noaa.gov/gmd/ccgg/trends_ch4/, viewed on 30/09/2016*

Page 8, section 2.2.6: Where does the MEGAN model implemented in this CTM derive the leaf-area index needed to calculate biogenic emissions?

*Response: The leaf-area index is obtained from the MEGANv2.04 databases (http://lar.wsu.edu/megan/guides.html). The data is originally at 150 sec horizontal resolution and it is averaged to the NMMB/BSC-CTM model grid. It is described in Guenther et al. (2006).*

*Guenther, A. et al. Estimates of global terrestrial isoprene emissions using MEGAN (Model of Emissions of Gases and Aerosols from Nature). Atmos. Chem. Phys 1–30 (2006).*

Page 8, line 29: Need a period after "parameters".
*Response: Amended.*

Page 9, section 3: What is the size of the bottom-most layer in the model? Also provide an estimate of the time it takes to run a year's simulation.
*Response: The size of the bottom-most layer in the model is below 40 m. This information is now included in the revised manuscript. The time to run a yearly simulation is about 2 weeks using 132 cores in the Marenostrum supercomputer based on Intel SandyBridge-EP E5-2670/1600 20M 8-core at 2.6 GHz.*

Page 9, lines 12-13: Since emissions after year 2000 were not provided by Lamarque et al., which projection (RCP?) was used for 2010 emissions to perform linear interpolation?
*Response: Thanks for pointing this out. There was an error in the description of the methodology used to derive the 2004 anthropogenic and biomass burning emissions. We stated that these emissions were obtained by interpolation between years 2000 and 2010. In reality we considered the emissions for year 2000 from Lamarque et al. (2010). This issue has been clarified in the revised manuscript, as follows: "Note that this methodology involves*

*assuming 2004 emissions equivalent to the best estimate reported for ACCMIP for year 2000".*

*Lamarque, J.-F., Bond, T. C., Eyring, V., Granier, C., Heil, A., Klimont, Z., Lee, D., Liousse, C., Mieville, A., Owen, B., Schultz, M. G., Shindell, D., Smith, S. J., Stehfest, E., Van Aardenne, J., Cooper, O. R., Kainuma, M., Mahowald, N., McConnell, J. R., Naik, V., Riahi, K., and van Vuuren, D. P.: Historical (1850–2000) gridded anthropogenic and biomass burning emissions of reactive gases and aerosols: methodology and application, Atmos. Chem. Phys., 10, 7017-7039, doi:10.5194/acp-10-7017-2010, 2010.*

Page 10, line 25: Any particular reason why only two aircraft campaigns were used for the evaluation instead of several others available in Emmons et al. (2000).

*Response: The model was evaluated against all the campaigns available in Emmons et al. (2000). However, for the paper we selected the two closest campaigns to year 2004. Figure 1 shows the comparison of the model with the PEM-Tropics and POLINAT-2 campaigns for HNO3, NOx and PAN. We have included Figure 1 in the supplementary material (see Figure S5) and additional text describing the results in the main manuscript (see also Table 5).*

[Figure]

*Figure 1. Comparison of modeled (black lines) and observed (red lines) vertical profiles of NOx and HNO3 and PAN for Tahiti and Ireland. Horizontal lines show the standard deviations.*

*Emmons, L. K., Hauglustaine, D. A., Müller, J.-F., Carroll, M. A., Brasseur, G. P., Brunner, D., Staehelin, J., Thouret, V., and Marenco, A.: Data composites of airborne observations of tropospheric ozone and its precursors, Journal of Geophysical Research: Atmospheres, 105, 20 497–20 538, doi:10.1029/2000JD900232, http://dx.doi.org/10.1029/2000JD900232, 2000.*

Page 12, section 5.1: What is the simulated tropospheric lifetime of methane in the model and how does it compare with that from multi-model studies (e.g., Naik et al 2013a)?

*Response: Methane lifetime was not explicitly calculated during model execution. While the burden can be calculated in post processing the estimation of the mean tropospheric methane – OH oxidation flux (needed to calculate lifetime) would require repeating the simulations. Therefore, we can neither include this information, nor discuss it at present in the manuscript.*

How does the simulated OH interhemispheric ratio compare with other studies (Naik et al., 2013a; Patra et al., 2014).

*Response: The mean OH inter-hemispheric (N/S) ratio of the model is 1.18. This quantity is comparable with the present-day multi-model mean ratio (1.28 ± 0.1) shown in Naik et al., (2013). This information is now included in the revised manuscript.*

*Naik, V., Voulgarakis, A., Fiore, A. M., Horowitz, L. W., Lamarque, J.-F., Lin, M., Prather, M. J., Young, P. J., Bergmann, D., Cameron-Smith, P. J., Cionni, I., Collins, W. J., Dalsøren, S. B., Doherty, R., Eyring, V., Faluvegi, G., Folberth, G. A., Josse, B., Lee, Y. H., MacKenzie, I. A., Nagashima, T., van Noije, T. P. C., Plummer, D. A., Righi, M., Rumbold, S. T., Skeie, R., Shindell, D. T., Stevenson, D. S., Strode, S., Sudo, K., Szopa, S., and Zeng, G.: Preindustrial to present-day changes in tropospheric hydroxyl radical and methane lifetime from the Atmospheric Chemistry and Climate Model Intercomparison Project (ACCMIP), Atmos. Chem. Phys., 13, 5277-5298, doi:10.5194/acp-13-5277-2013, 2013.*

Lightning NOx emissions have been shown to contribute significantly to tropospheric OH concentrations (Murray et al., 2013). Lightning NOx emissions are not considered in the current model set-up. Please explain how the simulated OH concentrations match closely with those of other modeling studies that include lightning NOx emissions.

*Response: We calculated the regional mean air mass-weighted OH concentrations and they are close to the multi-model values in Naik et al. (2013a) (see Fig. 2). Over the tropics (30S-30N) our OH is slightly higher, and above 500 hPa, is lower than the multi-model mean. Labrador et al. (2004) studied the sensitivity of OH to NOx from lightning. They showed that OH increases mostly in the middle to upper troposphere (500-200 hPa) when lightning emissions are considered. Accordingly, the lack of lightning emissions in our model could explain the lower OH values above 500 hPa reported here. This discussion is now included in the revised manuscript and Fig. 2 (right panel) is included in the supplementary material as Figure S2.*

[Figure]

[Figure]

| Naik et al. 2013 | NMMB/BSC-CTM |

*Figure 2. Comparison between the model and Naik et al., (2013) regional mean airmass-weighted OH concentrations ($\times 10^5$ molecule cm$^{-3}$).*

*Labrador, L. J., R. von Kuhlmann, and M. G. Lawrence (2004), Strong sensitivity of the global mean OH concentration and the tropospheric oxidizing efficiency to the source of NOx from lightning, Geophys. Res. Lett., 31, L06102, doi:10.1029/2003GL019229*

*Naik, V., Voulgarakis, A., Fiore, A. M., Horowitz, L. W., Lamarque, J.-F., Lin, M., Prather, M. J., Young, P. J., Bergmann, D., Cameron-Smith, P. J., Cionni, I., Collins, W. J., Dalsøren, S. B., Doherty, R., Eyring, V., Faluvegi, G., Folberth, G. A., Josse, B., Lee, Y. H., MacKenzie, I. A., Nagashima, T., van Noije, T. P. C., Plummer, D. A., Righi, M., Rumbold, S. T., Skeie, R., Shindell, D. T., Stevenson, D. S., Strode, S., Sudo, K., Szopa, S., and Zeng, G.: Preindustrial to present-day changes in tropospheric hydroxyl radical and methane lifetime from the Atmospheric Chemistry and Climate Model Intercomparison Project (ACCMIP), Atmos. Chem. Phys., 13, 5277-5298, doi:10.5194/acp-13-5277-2013, 2013.*

Page 12, line 28: Need a reference for aerosol influence on OH. Also, a larger oxidizing capacity would be simulated if lightning NOx emissions were included in these simulations.

*Response: Real and Sartelet (2011) studied the effect of aerosols in the photolysis rates and gaseous species, showing that differences in photolysis rates lead to changes in gas concentrations, with the largest impact simulated on OH and NO concentrations. At the ground, monthly mean concentrations of both species were reduced over Europe by around 10 to 14% and their tropospheric burden by around 10%. The decrease in OH led to an increase of the lifetime of several species such as VOC. On the other hand, Bian et al. (2003) evaluated the effect of aerosols on the global budgets of O3, OH and CH4 through their alteration of photolysis rates. The impact identified was to increase tropospheric O3 by 0.63 Dobson units and increase tropospheric CH4 by 130 ppb (via tropospheric OH decreases of 8%). Although the CH4 increases were global, the changes in tropospheric OH and O3 were mainly regional, with the largest impacts in northwest Africa for January and in India and southern Africa for July.*

*As we have described in a previous comment, a larger oxidizing capacity would be simulated, especially above 500 hPa, if lightning NOx emissions were included in our model run (Labrador et al., 2004).*

*Both aspects, and corresponding references, are now discussed in the revised version of the manuscript with the following paragraph: "Therefore, the lack of lightning emissions in our model run could at least partly explain the lower OH values above 500 hPa reported here. Another potential explanation is the lack of aerosols in our simulation, which may overestimate photolysis rates in polluted regions (e.g., Bian et al., 2003; Real and Sartelet, 2011)."*

*Real, E., and K. Sartelet. "Modeling of photolysis rates over Europe: impact on chemical gaseous species and aerosols." Atmospheric Chemistry and Physics 11.4 (2011): 1711-1727.*

*Bian, H., M. J. Prather, and T. Takemura, Tropospheric aerosol impacts on trace gas budgets through photolysis, J. Geophys. Res., 108(D8), 4242, doi:10.1029/2002JD002743, 2003.*

*Labrador, L. J., R. von Kuhlmann, and M. G. Lawrence (2004), Strong sensitivity of the global mean OH concentration and the tropospheric oxidizing efficiency to the source of NOx from lightning, Geophys. Res. Lett., 31, L06102, doi:10.1029/2003GL019229*

Page 13, line 9-10: Please give the reason why there were low CO concentrations in 2004 despite large Alaskan and Canadian wildfires.

*Response: Elguindi et al., 2010 presented a global analysis of observed CO seasonal averages and interannual variability for the years 2002-2007. They analyzed the CO concentrations during this period: "In JJA 2003, the anomalously high concentrations of CO due to the intense heat wave experienced in Europe, especially in August (Tressol et al.,2008; Ordoñez et al.,2010), are well represented in the data. Likewise, the high concentrations seen in SON 2002 are due to exceptional circumstances, namely the intense boreal forest fires which occurred over western Russia (Edwards et al.,2004; Yurganov et al., 2005; Kasischke et al.,2005)."*

*In summary, there were also important fires during the period 2002-2007 and meteorological conditions that could have an impact to the CO concentrations, like the intense heat wave or the photochemical conditions. This is why Elguindi et al., 2010 concluded that "despite the intense boreal forest fires that occurred during the summer in Alaska and Canada, the year 2004 had comparably lower tropospheric CO concentrations".*

*Elguindi, N., Clark, H., Ordóñez, C., Thouret, V., Flemming, J., Stein, O., Huijnen, V., Moinat, P., Inness, A., Peuch, V.-H., Stohl, A., Turquety, S., Athier, G., Cammas, J.-P., and Schultz, M.: Current status of the ability of the GEMS/MACC models to reproduce the tropospheric CO vertical distribution as measured by MOZAIC, Geoscientific Model Development, 3, 501–518, doi:10.5194/gmd-3-501-2010, http://www.geosci-model-dev.net/3/501/2010/, 2010.*

Page 13, 11-15: Other modeling studies suggest even lower CO burden (e.g., Naik et al., 2013b). Could higher CH4 concentration prescribed in the model play a role in the simulated high CO burden?

*Response: The influence of CH4 on CO has been assessed through a short sensitivity test. Changing the CH4 prescribed value from 1.85 ppm (NH background average) to 1.78 ppm (global average for 2004) lead to changes in daily average CO concentration up to ±0.12 ppb, which leads us to believe that other factors have a larger impact on CO burden (see for instance Shindell et al., 2006).*

*Shindell, D. T., et al. (2006), Multimodel simulations of carbon monoxide: Comparison with observations and projected near-future changes, J. Geophys. Res., 111, D19306, doi:10.1029/2006JD007100*

Page 13, last paragraph: the role of seasonal CO emissions in explaining the low northern hemisphere wintertime bias has been highlighted by Stein et al., (2014), which should be noted here.

*Response: Stein et al., (2014) is already discussed in the manuscript on page 14 and 15. On page 14 of the manuscript "During winter and spring, Stein et al. (2014) also obtain an underestimation of CO vertical profiles in airports located in the Northern Hemisphere (NH)" and "The wintertime negative bias (~ - 10-35 ppb) in the NH may be explained by either the lack of seasonally varying anthropogenic emissions in our simulation, an underestimation of CO emissions (Stein et al, 2014), or a combination thereof". On page 15 "Stein et al. (2014) suggests that the persistent negative bias in northern mid-latitude CO in models is most likely due to a combination of too low road traffic emissions and dry deposition errors."*

*Stein et al., (2014) On the wintertime low bias of Northern Hemisphere carbon monoxide found in global model simulations, Atmos. Chem. Phys., 14, 9295-9316, doi:10.5194/acp-14-9295-2014.*

Page 15, line 25: Give the lifetime of NOx.

*Response: The lifetime of NOx varies considerably with altitude, being only a few hours near the PBL and up to a few days in the upper troposphere (Tie et al., 2001 and 2002). This information is now included in the revised manuscript with the following sentence: "It has a relatively short lifetime (a few hours near the PBL and up to a few days in the upper troposphere; Tie et al., 2001 and 2002)".*

*Tie, X., R. Zhang, G. Brasseur, L. Emmons, and W. Lei (2001), Effects of lightning on reactive nitrogen and nitrogen reservoir species in the troposphere, J. Geophys. Res., 106(D3), 3167–3178, doi:10.1029/2000JD900565.*
*Tie, X., Zhang, R., Brasseur, G. et al. Journal of Atmospheric Chemistry (2002) 43: 61. doi:10.1023/A:1016145719608*

Page 18, line 13-18: How do the calculated dry deposition estimates compare with those from more recent chemistry-climate model simulations (e.g., Naik et al., 2013b).

*Response: The calculated dry deposition (1209 Tg O3) is higher than in TM5 (829 Tg O3) and MOZART-2 (857 Tg O3) and similar to LMDz-INCA (1261 Tg O3) and the multimodel ensemble in Stevenson et al. (2006) (1003 -+ 200 Tg O3). In addition, the model shows similar results to the GFDL AM3 chemistry-climate model (Naik et al., 2013b). The reference to GFDL AM3 model is now added in the revised manuscript.*

*Naik et al., 2013b, Impact of preindustrial to present day changes in short-lived pollutant emissions on atmospheric composition and climate forcing, J. Geophys. Res., doi: 10.1002/jgrd.50608.*

Page 19: What fraction of model O3 biases could be related to biases in the simulated meteorological fields (e.g., temperature)?

*Response: This is not an easy question to answer. Ozone is sensitive to temperature, solar radiation and vertical mixing. It is clear that biases in the meteorology will have a significant impact on the ozone biases. Another study would be required to provide a thorough quantification of the biases and this is beyond the scope of the present work. In any case, the NMMB meteorological skills are under constant improvement at NCEP and the authors consider the computed meteorology to lie within the skills of current state-of-the-art meteorological models.*

Map figures: Please remove the grey background on the maps as this makes it difficult to read the colours.
*Response: We think that the color scale is readable, according to the figure´s purpose, and if we remove the grey background the white dots in the figure would not be visible. Therefore, we kept the figures with the grey background.*

Figure 6, 10: Colour bar text is too small to read.
*Response: Amended.*

---

## Author Comment (AC3) · 11 Nov 2016

**RC2: 'Review of Badia et al. 2016', Anonymous Referee #2, 02 Aug 2016**

Review of "Gas-phase chemistry in the online multiscale NMMB/BSC Chemical Transport Model: Description and evaluation at global scale" by Badia et al.

Overview:

The paper is a description and an evaluation of a one-year (2004) global simulation of what is called the NMMB/BSC Chemical Transport Model. The model results for CO, O3 and NO2/PAN/HNO3 are compared against surface observations, profile observations and satellite retrievals. The comparison shows that the NMMB/BSC model gives acceptable results but both CO and ozone are overestimated on the global scale.

General remarks:

My two main concerns with the presented model run for 2004 is (i) that NOx emissions from lightning were not considered in the model run and (ii) that no specific biomass burning emissions for 2004 were used. A parameterisation of lightning emissions is scientific standard in global CTMs and there is no good reason, why such an important contribution to global tropospheric chemistry can be omitted. Getting daily or 8-daymean 2004 biomass emission data (GFED, GFAS etc.) would not have been difficult. Also the lack of seasonality of the anthropogenic emissions is an unnecessary simplification. Against the backdrop of these omissions it becomes difficult to draw conclusion from the model results and it severely undermines the scientific credibility of the paper.

*Response: The authors wish to thank anonymous reviewer #2 for the valuable comments and suggestions.*

*Note that we have decided to rename our model following a comment from reviewer #2 about avoiding the use of CTM for an online model. Thus, the new name is NMMB-MONARCH, where MONARCH stands for "Multiscale Online Nonhydrostatic AtmospheRe CHemistry model". In the responses to the reviewer's comments we keep the NMMB/BSC-CTM name to keep consistency with the manuscript submitted to GMDD, but in the revised manuscript the new name, NMMB-MONARCH, is used.*

*Now, the revised manuscript is entitled "Description and evaluation of the Multiscale Online Nonhydrostatic AtmospheRe CHemistry model (NMMB-MONARCH) version 1.0: gas-phase chemistry at global scale".*

*We agree with the reviewer that the model run presented in the manuscript has some shortcomings. However, we disagree that they undermine the scientific credibility of the paper. This contribution describes a first major step in the development of a new multiscale chemical weather prediction system and its thorough evaluation, all of which is within the scope of GMD. Improvements to the system are conducted on a regular basis in our group, and the inclusion of NOx emissions from lightning is planned for the next version of the model. The selection of the emissions was an intentional decision. Not only the thorough evaluation with observations but also the comparison to other scientific studies was our priority. Considering the timeline of the Atmospheric Chemistry and Climate Model Intercomparison Project (ACCMIP) initiative, we made use of the emissions presented in*

*Lamarque et al. (2010). Thus, we took advantage of this excellent opportunity to evaluate the NMMB/BSC-CTM model in a way that can be consistently compared to other modeling systems contributing to ACCMIP. In this sense, emissions representative of the 2000 decade were used both for anthropogenic and biomass burning sources. This decision implied some drawbacks in our work, such as the non specificity of the emissions for year 2004, or the lack of seasonal variability on anthropogenic emissions. We consider those limitations as minor compared to the benefits of comparing our simulations to those of the ACCMIP experiment. The selected year of simulation provides a reference to identify the skills and limitations of the methods applied, but it is not intended to reproduce exactly the same episodes observed during 2004. Our goal is to reproduce the annual trends and patterns of the main atmospheric chemistry components. In the manuscript, all the limitations of our simulations are identified and described. In the revised manuscript version, we have provided additional discussion on these issues along with their implications. For example, the lack of lightning emissions helps explaining the underestimation of OH in the middle to upper troposphere. The results of this evaluation and comparison have allowed us identifying the next steps that will be required to improve the modeling system.*

*The lack of seasonality in the anthropogenic emissions is considered a minor limitation. Currently, there are still important uncertainties on the estimation of anthropogenic emissions at global scale, and most of the available modeling studies of atmospheric chemistry at global scales run the experiments with constant profiles. This can be explained by the resolution of study, as models are configured at global horizontal resolutions of a few degrees. Under such conditions, the daily variability of the emissions may not be the most important issue to address. Regarding biomass burning emissions, Marlier et al. (2014) showed that going from monthly to daily fire emissions does not change the atmospheric composition very drastically, especially for gases. This supports our initial approach of not using specific daily emissions of biomass burning for 2004.*

*We believe that the constructive discussion of the model's behavior in the absence of the aforementioned processes can provide useful insight into the importance of those processes.*

*Lamarque, J.-F., Bond, T. C., Eyring, V., Granier, C., Heil, A., Klimont, Z., Lee, D., Liousse, C., Mieville, A., Owen, B., Schultz, M. G., Shindell, D., Smith, S. J., Stehfest, E., Van Aardenne, J., Cooper, O. R., Kainuma, M., Mahowald, N., McConnell, J. R., Naik, V., Riahi, K., and van Vuuren, D. P.: Historical (1850–2000) gridded anthropogenic and biomass burning emissions of reactive gases and aerosols: methodology and application, Atmos. Chem. Phys., 10, 7017-7039, doi:10.5194/acp-10-7017-2010, 2010.*
*Marlier, M.E., A. Voulgarakis, D.T. Shindell, G. Faluvegi, C.L. Henry, and J.T. Randerson, 2014: The role of temporal evolution in modeling atmospheric emissions from tropical fires. Atmos. Environ., 89, 158-168, doi:10.1016/j.atmosenv.2014.02.039.*

On the other hand, the presented model had two advanced properties, namely the online calculation of VOC emissions using the MEGAN model and the fact that the presented model is an on-line coupled chemistry – meteorological model (or ChemistryGCM). The term CTM, which the authors choose, is commonly used for off-line model without the simulation of meteorology (see Baklanov et al. 2014, ACP). I therefore recommend not to use the term CTM in the name of the model because it is an on-line coupled model. Unfortunately, it is a missed chance that these two new aspects were not explored further in the paper.

*Response: This manuscript represents our first step towards the development of a fully coupled chemistry-meteorology model. We intend to present the current state of development of the model (version 1.0) with a comprehensive evaluation using a wide variety of observations. This will serve as a baseline for further studies where more complexity will be added to the system and where specific sensitivity studies may be conducted. Future work will analyze the impact of the online nature of the model, and to assess the sensitivity to the online calculation of biogenic emissions.*

*Concerning the name of the model (NMMB/BSC-CTM), we have considered the suggestion of the reviewer to not use the term CTM (Chemistry Transport Model) in the model name. CTM has been used traditionally for offline chemistry models, and we agree with the reviewer that using it may lead to some confusion. Considering the nature of our model, a pure online meteorology-chemistry system, we have decided to completely rename the model. The new name is NMMB-MONARCH, where MONARCH stands for "Multiscale Online Nonhydrostatic Atmosphere Chemistry model". We believe that with the new name, all the major characteristics of the system are clearly stated. Now in the revised manuscript all the references to NMMB/BSC-CTM have been replaced with NMMB-MONARCH. We only keep a reference to the old name in the abstract and introduction section to link the previous developments with the new naming adopted. Thus, the title of the manuscript is now "Description and evaluation of the Multiscale Online Nonhydrostatic AtmospheRe CHemistry model (NMMB-MONARCH) version 1.0: gas-phase chemistry at global scale".*

The evaluation is carried out with a well-balanced choice of observations but the results are too often only described with the words such as "good agreement" etc. I think this is not very meaningful, instead the results should be quantified in a better way, i.e. a bias of 10 ppb, 20% etc.

*Response: The results are now better quantified in the revised manuscript, and although expressions as "good agreement" are maintained, they come with a quantification of statistics to support them.*

It is a thought-provoking result that both CO and ozone are overestimated because an overestimation of the oxidation capacity is often linked with CO underestimation (see Strode et al. 2015, ACP) It is something which can not be found in other models using similar emission data, and especially for models that also use the CB05 chemical mechanism. I think this result deserves a more thorough investigation.

*Response: In our case, we attribute the CO overestimation mainly to emission sources. Even, if some other models using similar emissions don't show this overestimation, we have detected significant sensitivity of the biogenic emissions from MEGAN (as discussed in the manuscript) to the temporal basis of the meteorological conditions. This could affect CO emission overestimation.*

*On the other hand, the O3 overestimation can be due to multiple factors, such as the lack of halogen and heterogeneous chemistry, an overestimation of the STE for some seasons, the lack of aerosols in the simulation, uncertainties in the solar radiation reproduced by the model, the lack of seasonal variation on anthropogenic emissions, and/or deviations in dry/wet deposition rates.*

*It is out of the scope of this work to develop specific sensitivity analysis to analyze the influence of each of those factors (as in Strode et al., 2015). Future works will be devoted specifically to these tasks.*

Without carrying out sensitivity studies, it is in general problematic to come to valid conclusion on the reasons for certain aspects (bad or good) of the model performance. The authors predominately only argue (without doing sensitivity studies) that (i) defi- ciency in the emissions and/or (ii) the lack of considering aerosol in the photolysis rates are the reasons for identified model deficiencies. While there is consensus in the scientific community that emissions can be very uncertain, there is no evidence given in the paper, why the aerosol impact should be so important as the authors claim. (I am happy to be convinced otherwise by a sensitivity study or a reference to it).

*Response: The authors agree that the emphasis put during the discussion on the uncertainty of emissions and the lack of aerosols in the previous version of the paper could lead to the misconception that we believe those are the only causes for the disagreements found between model and observations. The revised manuscript has been thoroughly revised to point out other known causes for poor model performance, which have been referenced in all cases.*

*For ozone, other factors contributing to the deviations could be the lack of halogen chemistry in the CB05 mechanism, an excess of STE towards the troposphere, or possible bias in the meteorology (solar radiation, temperature). Sherwen et al. (2016) show that the halogen chemistry reduces the global tropospheric ozone burden by 15%. On the other hand, Real and Sartelet (2011) studied the effect of aerosols on photolysis rates and gaseous species and found that "Differences in photolysis rates lead to changes in gas concentrations, with the largest impact simulated on OH and NO concentrations. At the ground, monthly mean concentrations of both species are reduced over Europe by around 10 to 14% and their tropospheric burden by around 10%. The decrease in OH leads to an increase of the lifetime of several species such as VOC". Additionally, Bian et al. (2003) evaluated the effect of aerosols on the global budgets of O3, OH and CH4 through their alteration of photolysis rates. The impact identified was to increase tropospheric O3 by 0.63 Dobson units and increase tropospheric CH4 by 130 ppb (via tropospheric OH decreases of 8%). Although the CH4 increases were global, the changes in tropospheric OH and O3 were mainly regional, with the largest impacts in northwest Africa for January and in India and southern Africa for July. These last works, supports the idea introduced in our manuscript that the role of aerosols and photolysis is also important. In the revised manuscript, we have included these possible causes for the ozone bias discussion.*

*Real, E. and Sartelet, K.: Modeling of photolysis rates over Europe: impact on chemical gaseous species and aerosols, Atmos. Chem. Phys., 11, 1711-1727, doi:10.5194/acp-11-1711-2011, 2011.*

*Bian, H., M. J. Prather, and T. Takemura, Tropospheric aerosol impacts on trace gas budgets through photolysis, J. Geophys. Res., 108(D8), 4242, doi:10.1029/2002JD002743, 2003.*

*Sherwen, T., Schmidt, J. A., Evans, M. J., Carpenter, L. J., Großmann, K., Eastham, S. D., Jacob, D. J., Dix, B., Koenig, T. K., Sinreich, R., Ortega, I., Volkamer, R., Saiz-Lopez, A., Prados-Roman, C., Mahajan, A. S., and Ordóñez, C.: Global impacts of tropospheric halogens (Cl, Br, I) on oxidants and composition in GEOS-Chem, Atmos. Chem. Phys. Discuss., doi:10.5194/acp-2016-424, in review, 2016.*

The authors should discuss other aspect of their model setup in more detail. If there is the feeling that photolysis rates play a role, then cloud cover would be the first suspect. The

cloud cover should be checked for biases since the model simulates clouds itself. Also worth checking are the ozone total columns used in the photolysis scheme because they are not constrained by observations.

*Response: This manuscript aims to describe the tropospheric gas-phase chemistry of a new modeling system. Although, the comment of the reviewer asking for information about realism of the stratospheric ozone and clouds is relevant, we believe its discussion goes beyond the scope of the paper, and we have decided not to explicitly include it in the revised manuscript. However, we provide below information about the goodness of both simulated fields. In the revised manuscript we introduce the role of STE in the O3 biases.*

*The stratospheric ozone in the NMMB/BSC-CTM can be computed with linear models, on one hand the COPCAT scheme and on the other, as an alternative, the Cariolle scheme. In the PhD of Dr. Alba Badia (Badia, 2014), both systems were evaluated against satellite data. Relevant biases on the COPCAT stratospheric ozone are:*

1) *COPCAT underestimates the stratospheric ozone maximum between 50N-20S latitudes and the ozone above 10 hPa. This underestimation is consistent with the study by Monge-Sanz et al. (2011), which compares the annual average from the year 2000 ozone profiles between the COPCAT parameterization and the HALOE measurements in the tropics (4 N).*

2) *COPCAT also underestimate the ozone in the low stratosphere between 90N-50N during DJF and MAM.*

3) *Over high southern latitudes COPCAT tends to overestimate O3 concentrations during the months of DJF and MAM in the mid/low stratosphere below 20hPa.*

*In addition, Figure 1 shows the comparison of the ozone total column computed with Cariolle and COPCAT and with SCIAMACHY satellite retrievals. Additionally, the work of Monge-Sanz et al. (2011) further describes the biases associated to the COPCAT linear model. It is important to note that COPCAT is used as the linear ozone scheme of the ECMWF global model system IFS.*

[Figure]

*Figure 1. Comparison of the ozone total column monthly mean computed with NMMB/BSC-CTM using Cariolle scheme (CAR) and COPCAT scheme (COP), and compared with the SCIAMACHY satellite retrievals.*

*Regarding the clouds and possible biases. Clouds are simulated by the NMMB meteorological model. The model has been and is being extensively evaluated by NCEP in its both global and limited area configurations. Results of the skills of NMMB at global scale have widely been presented at several international conferences (e.g. AGU, EGU General Assembly). Comparisons of the NMMB initialized with GFS analysis show competitive results of the NMMB compared with GFS. For example, in Figure 2 the anomaly correlation at day 5-forecast shows excellent results with NMMB initialized with GFS or ECMWF analysis. Better results are obtained with ECMWF analysis. Such results wouldn't be possible with a poor representation of the cloud cover, and provide clear information on the skill of the NMMB as a numerical weather forecast model at global scale.*

Anomaly correlation day-5 forecast

| North Hemisphere | | | | South Hemisphere | | | | |
|---|---|---|---|---|---|---|---|---|
| DATE | GFS | NMMB (GFS) | NMMB (ECMWF) | DATE | GFS | NMMB (GFS) | NMMB (ECMWF) | |
| 20110304 | 0.72 | 0.77 | 0.85 | 20110131 | 0.75 | 0.81 | 0.87 | |
| 20110317 | 0.71 | 0.79 | 0.89 | 20110209 | 0.75 | 0.77 | 0.88 | |
| 20110430 | 0.69 | 0.88 | 0.89 | 20110222 | 0.70 | 0.75 | 0.81 | |
| 20110616 | 0.75 | 0.84 | 0.83 | 20110306 | 0.65 | 0.69 | 0.84 | |
| 20110617 | 0.74 | 0.73 | 0.84 | 20110320 | 0.71 | 0.86 | 0.88 | |
| 20110618 | 0.73 | 0.81 | 0.88 | 20110325 | 0.62 | 0.69 | 0.70 | |
| 20110619 | 0.62 | 0.64 | 0.81 | 20110327 | 0.72 | 0.66 | 0.76 | |
| 20110620 | 0.70 | 0.61 | 0.72 | 20110403 | 0.75 | 0.64 | 0.74 | |
| 20110629 | 0.63 | 0.79 | 0.88 | 20110414 | 0.72 | 0.72 | 0.75 | |
| 20110702 | 0.63 | 0.90 | 0.94 | 20110421 | 0.75 | 0.78 | 0.90 | |
| 20110713 | 0.71 | 0.38 | 0.55 | 20110506 | 0.68 | 0.66 | 0.90 | |
| 20110718 | 0.68 | 0.59 | 0.58 | 20110507 | 0.72 | 0.84 | 0.88 | |
| 20110806 | 0.72 | 0.83 | 0.84 | 20110725 | 0.75 | 0.88 | 0.90 | |
| 20110807 | 0.71 | 0.73 | 0.83 | 20110728 | 0.72 | 0.88 | 0.90 | |
| 20110808 | 0.73 | 0.87 | 0.91 | 20110925 | 0.73 | 0.73 | 0.84 | |
| 20110813 | 0.69 | 0.84 | 0.85 | | | | | |
| 20110912 | 0.71 | 0.76 | 0.81 | | | | | |

•11/32 significant improvement with both analyses
•12/32 significant improvement with ECMWF analysis in comparison with run with GFS analysis
•17/32 improvement in comparison to GFS with same analysis
•2/32 score remain less then then 0.7 with both analyses
•On average better scores with ECMWF analyses

22/09/16          Z. Janjic          33

*Figure 2. Anomaly correlation day-5 forecast in the Northern and Southern Hemisphere obtained with the NMMB global model initialized with GFS and ECMWF analysis. Days selected when GFS forecast was poor. (Source: Z. Janjic personal communication).*

*However, some works have shown minor tropospheric effects of clouds on ozone (e.g., Voulgarakis et al., 2009; Liu et al., 2006). This points to that clouds are unlikely to have a large global effect on tropospheric gas-phase chemistry (only regionally).*

*Badia i Moragas, A., 2014. Implementation, development and evaluation of the gas-phase chemistry within the Global/Regional NMMB/BSC Chemical Transport Model (NMMB/BSC-CTM). PhD Dissertation, Universitat Politècnica de Catalunya.*

*Monge-Sanz, B. M., Chipperfield, M. P., Cariolle, D., and Feng, W.: Results from a new linear O3 scheme with embedded heterogeneous chemistry compared with the parent full-chemistry 3-D CTM, Atmos. Chem. Phys., 11, 1227-1242, doi:10.5194/acp-11-1227-2011, 2011.*

*Voulgarakis, A., Wild, O., Savage, N. H., Carver, G. D., and Pyle, J. A.: Clouds, photolysis and regional tropospheric ozone budgets, Atmos. Chem. Phys., 9, 8235-8246, doi:10.5194/acp-9-8235-2009, 2009.*

*Liu, H., et al. (2006), Radiative effect of clouds on tropospheric chemistry in a global three-dimensional chemical transport model, J. Geophys. Res., 111, D20303, doi:10.1029/2005JD006403.*

Another potentially important aspect is the fact the emissions are injected uniformly in the lowest 500m (anthropogenic) or 1300 m (biomass burning). This could have a large impact on dry deposition, which depends on the surface level concentration, and ozone titration by NO during the night. The 500 m seems to be an exaggeration of the extent of the mixed layer during the night over land and the choice needs to be better motivated. One would expect that the diffusion scheme of the model simulates the vertical mixing in the PBL. Also, the 1300 m for the biomass burning injection would need to be justified, as the fire injection height can vary substantially (see for example. Remy al., 2016, ACP).

*Response: The authors agree that the injection of emissions in global models is a critical point. However, there is no clear consensus within the modeling community in the approaches to be applied, as can be seen in different works where different approaches are applied from vertically distributing the emissions in height, injecting the emissions in the first model layers, or simply in the first model layer (i.e., Emmons et al., 2010, Huijnen et al., 2010). In our case, we consider that a global model with 1° of horizontal resolution and 64 vertical layers has limitations in the vertical diffusion of pollutants injected in the surface layer due to the lack of detail in the type of landuse or soil properties at specific locations. The coarse resolution strongly affects the surface concentration and we found beneficial to inject the emissions in the PBL for anthropogenic and biomass burning. In the case of anthropogenic emissions, we defined a constant injection height of 500 m that will distribute the emissions within the convective boundary layer during the day, and will trap the emissions in a stable boundary layer during nighttime. With higher resolutions this first approach might be detrimental to the model skills, and a more detailed injection of the emissions will be considered. Regarding biomass burning, the injection within the first 1300 m was considered a good approximation of injecting the emissions within the PBL and is within the range of emission heights recommended by Dentener (2006). Furthermore, some studies recommend an increase in the injection height in the tropics to 2 km based on the evidence from recent satellite observations (e.g. Labonne et al., 2007), which is a height much higher than that selected in our simulations.*

*Dentener, F., Kinne, S., Bond, T., Boucher, O., Cofala, J., Generoso, S., Ginoux, P., Gong, S., Hoelzemann, J. J., Ito, A., Marelli, L., Penner, J. E., Putaud, J.-P., Textor, C., Schulz, M., van der Werf, G. R., and Wilson, J.: Emissions of primary aerosol and precursor gases in the years 2000 and 1750 prescribed data-sets for AeroCom, Atmos. Chem. Phys., 6, 4321–4344, doi:10.5194/acp- 6-4321-2006, 2006b.*

*Emmons, L. K., Walters, S., Hess, P. G., Lamarque, J.-F., Pfister, G. G., Fillmore, D., Granier, C., Guenther, A., Kinnison, D., Laepple, T., Orlando, J., Tie, X., Tyndall, G., Wiedinmyer, C., Baughcum, S. L., and Kloster, S.: Description and evaluation of the Model for Ozone and Related chemical Tracers, version 4 (MOZART-4), Geosci. Model Dev., 3, 43-67, doi:10.5194/gmd-3-43-2010, 2010.*

*Huijnen, V., Williams, J., van Weele, M., van Noije, T., Krol, M., Dentener, F., Segers, A., Houweling, S., Peters, W., de Laat, J., Boersma, F., Bergamaschi, P., van Velthoven, P., Le Sager, P., Eskes, H., Alkemade, F., Scheele, R., Nédélec, P., and Pätz, H.-W.: The global chemistry transport model TM5: description and evaluation of the tropospheric chemistry version 3.0, Geosci. Model Dev., 3, 445-473, doi:10.5194/gmd-3-445-2010, 2010.*

*Labonne, M., Breon, F.-M., and Chevallier, F.: Injection heights of biomass burning aerosols as seen from a space borne lidar, Geophys. Res. Lett., 34, L11806, doi:10.1029/2007GL029311, 2007*

The paper would greatly benefit from proofreading for English language.
*Response: The final version of the manuscript has been revised in detail.*

Specific comments:

P1
Title: consider not calling the model a CTM as CTM's are understood as "off-line"
*Response: See the response to the general comment addressed above.*

P2
L 27: a better reference for IFS-MOZART is Flemming et al. 2009, GMD
*Response: Amended.*

L 33: Please clarify if the non-hydrostatic option was used in the run.
*Response: Yes, the nonhydrostatic option was turned on in the run although it is not necessary for the global scales used in our study. The extra computational cost of the nonhydrostatic dynamics is on the order of 10% in global applications, or nonexistent if the nonhydrostatic extension is switched off at coarser resolutions. However, the relatively low cost of the nonhydrostatic dynamics allows its application even at transitional resolutions where the benefits due to the nonhydrostatic dynamics are small or uncertain. See Janjic and Gall (2012) for further clarifications.*

*Janjic, Z. and Gall, R., 2012. Scientific documentation of the NCEP nonhydrostatic multiscale model on the B grid (NMMB). Part 1 Dynamics. NCAR/TN-489+ STR, 75 pp.*

P3
L 26: better "in detail"
*Response: Amended.*

P4
L15-25: please clarify which of the options is actually used in the presented run. The other options don't need to be mentioned. They could be referenced.
*Response: Table 1 presents the specific configuration of the NMMB used in this work. A comment regarding Table 1 is now introduced in the revised manuscript.*

P 5
L 17: 1850 ppb of methane seems too high for 2004. The value should be 1775 ppb http://www.esrl.noaa.gov/gmd/ccgg/trends_ch4/
*Response: In this first model version, we followed the approach to prescribe a constant methane concentration, which will be refined in future model versions. Current practices set the methane background level either as a default background concentration (i.e. 1.76 ppm, e.g. Shindell et al, 2006), or as the background level for the Northern Hemisphere (i.e. 1.85 ppm, used for instance within CMAQ). Including either of those concentrations would lead to differences with respect to reality and the choice in our case was made to include the value the closest to the present time global background concentration (1.83 ppm, see WMO 2015, or Dlugokencky, 2016), because we aim to apply this first version of the NMMB/BSC-CTM in forecast mode. The global average for 2004 is reported to be 0.06 to 0.07 ppm lower, around 4%.*

*Moreover, the sensitivity of the model results to the background methane concentration is relatively low. We have performed a test using a value of CH4 of 1.786 ppm (reported level by WMO for 2004) and compared it to the simulation with CH4 at 1.85 ppm. Differences in daily mean O3 and CO concentration up to ±0.6 ppb and ±0.12 ppb, respectively, are found for randomly selected days of April and August (Figure 3 depicts the differences for the 5th of August 2004 at surface level). The largest differences for O3 are not at the surface, but at higher layers (plots depict only surface level, but the maximum differences reported consider the full atmospheric column). In relative terms, differences can be considered low.*

[Figure]

*(a)*

[Figure]

*(b)*

*Figure 3. Difference in daily mean surface level concentration of (a) CO (ppb), and (b) O3 (ppb) for the 5th of August 2004, between a simulation with background methane concentration set to 1.85 ppm and another with 1.786 ppm.*

L 22: Add more information about the realism of the two input fields (overhead ozone and clouds)
*Response: This has already been addressed in the reply of a general comment above.*

P 6
L 10: Is this a monthly climatology ?
*Response: No. A specific value for season and landuse is provided.*

L 13: "cloud processes" – does this also include wet-phase chemistry ?
*Response: No. The version 1.0 of the model does not consider aqueous phase chemistry. This is now clarified in the revised manuscript.*

L 14: The presented terms are not clear. Please clarify what you mean by all the mentioned processes. For example, what is wet deposition for non-precipitating cloud?
*Response: The terms presented are common in the modeling context and are taken from Byun and Ching (1999) and Foley et al. (2010). The manuscript includes both references. The meaning of the terms used are:*
- *Grid-scale: it refers to those processes explicitly resolved by the model at the spatial scale of the model resolution.*
- *Subgrid-scale: it refers to those processes that are parameterized in the physics, i.e., convective clouds.*
- *Grid- scale Scavenging: it refers to the removal of gases by cloud droplets of clouds explicitly resolved by a numerical model, i.e., stratiform clouds.*
- *Grid-scale Wet deposition: it refers to the deposition by precipitation produced by explicitly resolved clouds.*
- *Subgrid-scale vertical mixing: it refers to the convective mixing that is produced within convective clouds.*
- *Subgrid-scale scavenging: it refers to the removal of gases by cloud droplets of clouds parameterized with a cumulus convection parameterization.*
- *Subgrid-scale wet deposition for precipitating: it refers to the deposition by precipitation produced by parameterized convective clouds.*
  *The relevant non-precipitating cloud processes are scavenging and vertical mixing. Wet deposition is possible for precipitating clouds. In order to avoid a too long section with common terminology definition the reader is referred to the scientific literature from where this terminology comes from.*

*Byun, D. and Ching, J.: Science algorithms of the EPA Models-3 community multiscale air quality (CMAQ) modeling system, Rep. EPA/600/R-99, 30, 1999.*

*Foley, K. M., Roselle, S. J., Appel, K. W., Bhave, P. V., Pleim, J. E., Otte, T. L., Mathur, R., Sarwar, G., Young, J. O., Gilliam, R. C., Nolte, C. G., Kelly, J. T., Gilliland, A. B., and Bash, J. O.: Incremental testing of the Community Multiscale Air Quality (CMAQ) modeling system version 4.7, Geoscientific Model Development, 3, 205–226, doi:10.5194/gmd-3-205-2010, http://www.geosci-model-dev.net/3/205/2010/, 2010.*

L16: Why only in-cloud scavenging and not all the other processes ?

*Response: The most relevant scavenging process affecting gases are the in-cloud scavenging. The American Meteorological Society glossary in its definition of scavenging by precipitation says "Rainout (or snowout), which is the in-cloud capture of particulates as condensation nuclei, is one form of scavenging. The other form is washout, the below-cloud capture of particulates and gaseous pollutants by falling raindrops. Large particles are most efficiently removed by washout. Small particles (especially those less than 1 μm in diameter) more easily follow the airstream flowing around raindrops and generally avoid capture by raindrops except in heavy rain events". In the case of gases, neglecting below-cloud scavenging is not considered a strong limitation.*

L 19: How is the cloud time scale derived?

*Response: Following Byun and Ching (1999) and Foley et al. (2010) the time scale of a convective cloud is assumed to be 3600s, and for explicitly resolved clouds, the model times step is applied.*

*Byun, D. and Ching, J.: Science algorithms of the EPA Models-3 community multiscale air quality (CMAQ) modeling system, Rep. EPA/600/R-99, 30, 1999.*
*Foley, K. M., Roselle, S. J., Appel, K. W., Bhave, P. V., Pleim, J. E., Otte, T. L., Mathur, R., Sarwar, G., Young, J. O., Gilliam, R. C., Nolte, C. G., Kelly, J. T., Gilliland, A. B., and Bash, J. O.: Incremental testing of the Community Multiscale Air Quality (CMAQ) modeling system version 4.7, Geoscientific Model Development, 3, 205–226, doi:10.5194/gmd-3-205-2010, http://www.geosci-model-dev.net/3/205/2010/, 2010.*

P7
L2: Do you refer to large-scale and convective precipitation here?

*Response: Yes, grid-scale refers to large-scale processes (those resolved by the grid information solved by the model) and subgrid-scale to convective processes.*

L7: "Convective mixing" do you mean transport by convective mass fluxes ?

*Response: Yes, convective mixing is referred to the processes that transport vertically within the convective cloud the air masses.*

L 13: 100 hPa is a rather high tropopause for mid- and high latitudes.

*Response: There is a common practice in modeling studies to implement stratospheric chemistry above 100 hPa (i.e., Lamarque et al., 2010; Voulgarakis et al., 2011). The chemistry in the tropopause is very slow and the transition within the tropospheric and stratospheric chemistry is wide. Considering the resolution of models in the upper troposphere and stratosphere makes the selection of 100 hPa a compromise solution.*

*Lamarque, J.-F., Bond, T. C., Eyring, V., Granier, C., Heil, A., Klimont, Z., Lee, D., Liousse, C., Mieville, A., Owen, B., Schultz, M. G., Shindell, D., Smith, S. J., Stehfest, E., Van Aardenne, J., Cooper, O. R., Kainuma, M., Mahowald, N., McConnell, J. R., Naik, V., Riahi, K., and van Vuuren, D. P.: Historical (1850–2000) gridded anthropogenic and biomass burning emissions of reactive gases and aerosols: methodology and application, Atmos. Chem. Phys., 10, 7017-7039, doi:10.5194/acp-10-7017-2010, 2010.*
*Voulgarakis, A., P. Hadjinicolaou, and J.A. Pyle, 2011: Increases in global tropospheric ozone following an El Niño event: Examining stratospheric ozone variability as a potential driver. Atmos. Sci. Lett., 12, 228-232, doi:10.1002/asl.318.*

L 13: Were these Mozart 4 fields evaluated for the stratosphere?

*Response: Yes. MOZART-4 model is described in Emmons et al., (2010) : "Mixing ratios of several species (O3, NOx, HNO3, N2O5, CO, CH4) are constrained in the stratosphere since MOZART-4 does not have complete stratospheric chemistry. These mixing ratios have been updated to zonal means from a MOZART-3 simulation." MOZART-3 is suitable for representing chemical/physical processes in stratosphere and for quantifying ozone fluxes from the stratosphere to the troposphere (Kinnison et al., 2007). The validity of STE processes in MOZART-3 has been evaluated in several studies (Park et al., 2004., Pan et al. 2007 and Liu et al., 2009). The model results in Park et al., 2004 showed good agreement for methane and water vapor, but underestimated the nitrogen oxide abundance.*

*Emmons, L. K., Walters, S., Hess, P. G., Lamarque, J.-F., Pfister, G. G., Fillmore, D., Granier, C., Guenther, A., Kinnison, D., Laepple, T., Orlando, J., Tie, X., Tyndall, G., Wiedinmyer, C., Baughcum, S. L., and Kloster, S.: Description and evaluation of the Model for Ozone and Related chemical Tracers, version 4 (MOZART-4), Geosci. Model Dev., 3, 43-67, doi:10.5194/gmd-3-43-2010, 2010.*

*Kinnison, D. E., et al. (2007), Sensitivity of chemical tracers to meteorological parameters in the MOZART-3 chemical transport model, J. Geophys. Res. , 112, D20302, doi:10.1029/2006JD007879.*

*Liu, Y., Liu, C. X., Wang, H. P., Tie, X. X., Gao, S. T., Kinnison, D., and Brasseur, G.: Atmospheric tracers during the 2003–2004 stratospheric warming event and impact of ozone intrusions in the troposphere, Atmos. Chem. Phys., 9, 2157-2170, doi:10.5194/acp-9-2157-2009, 2009.*

*Park, M., Randel, W. J., Kinnison, D. E., Garcia, R. R., and Choi, W.: Seasonal variation of methane, water vapor, and nitrogen oxides near the tropopause: Satellite observations and model simulations, J. Geophys. Res., 109, D03302, doi:10.1029/2003JD003706, 2004.*

*Pan, L. L., Wei, J. C., Kinnison, D. E., Garcia, R. R., Wuebbles, D. J., and Brasseur, G. P.: A set of diagnostics for evaluating chemistry-climate models in the extratropical tropopause region, J. Geophys. Res., 112, D09316, doi: 10.1029/2006JD007792, 2007.*

L 18ff: There is no need to present the COPCAT scheme here, a reference to the paper is enough.

*Response: Following the reviewer's comment, the description of the COPCAT linear scheme is removed from the manuscript and a reference, Monge-Sanz et al. (2011), is provided.*

*Monge-Sanz, B. M., Chipperfield, M. P., Cariolle, D., and Feng, W.: Results from a new linear O3 scheme with embedded heterogeneous chemistry compared with the parent full-chemistry 3-D CTM, Atmos. Chem. Phys., 11, 1227-1242, doi:10.5194/acp-11-1227-2011, 2011.*

L 18: Please provide information on the biases of your stratosphere ozone simulated by the COPCAT scheme because they have an impact on the photolysis rates.

*Response: As already discussed in the general comments, information on the stratospheric ozone simulated by COPCAT scheme can be found in the PhD of Dr. Alba Badia (Badia, 2014). From there, the most relevant biases on the stratospheric ozone identified are:*

1) *COPCAT underestimates the stratospheric ozone maximum between 50N-20S latitudes and the ozone above 10 hPa. This underestimation is consistent with the study by Monge-Sanz et al. (2011), which compares the annual average from the year 2000 ozone profiles between the COPCAT parameterization and the HALOE measurements in the tropics (4 N).*

2) *COPCAT also underestimate the ozone in the low stratosphere between 90N-50N during DJF and MAM.*

3) *Over high southern latitudes COPCAT tends to overestimate O3 concentrations during the months of DJF and MAM in the mid/low stratosphere below 20hPa.*

*In addition, Figure 1 shows the comparison of the ozone total column computed with Cariolle and COPCAT and with SCIAMACHY satellite retrievals. Additionally, the work of Monge-Sanz et al. (2011) further describes the biases associated to the COPCAT linear model. It is important to note that COPCAT is used as the linear ozone scheme of the ECMWF global model system.*

*In the revised manuscript the following sentence has been added "For further description of the approach and information on the biases of the stratosphere ozone simulated by the COPCAT scheme, the reader is referred to Monge-Sanz et al. (2011). "*

*Badia i Moragas, A., 2014. Implementation, development and evaluation of the gas-phase chemistry within the Global/Regional NMMB/BSC Chemical Transport Model (NMMB/BSC-CTM). PhD Dissertation, Universitat Politècnica de Catalunya.*

*Monge-Sanz, B. M., Chipperfield, M. P., Cariolle, D., and Feng, W.: Results from a new linear O3 scheme with embedded heterogeneous chemistry compared with the parent full-chemistry 3-D CTM, Atmos. Chem. Phys., 11, 1227-1242, doi:10.5194/acp-11-1227-2011, 2011.*

P 8

L 14: No lightning emissions is a severe shortcoming of the simulation and the paper (see my general comment).

*Response: This has been highlighted in the revised manuscript. The following sentence has been included "The omission of lightning emissions is expected to have a significant impact in the oxidation of the middle and upper troposphere" in Section 2.2.6.*

L 15: Please explain in more detail how the MEGAN code was integrated in your model.

*Response: The authors do not fully understand the comment of the reviewer regarding the coupling of MEGAN code with the NMMB/BSC-CTM. The MEGAN code has been prepared as a subroutine that is called by the chemistry driver of the model to compute an update on the biogenic emissions. The coupling is fully on-line integrated, so the meteorological variables (temperature and solar radiation) is passed as attribute to the subroutine. The authors consider that this explanation is not required in the manuscript.*

L34: I don't understand the 24h averages here. I thought (L23) the actual hourly meteorological data were used for the calculation of the Megan emissions.

*Response: The MEGAN model uses two different pieces of information regarding meteorological conditions. On one hand the model uses the actual temperature and solar radiation provided by the meteorological model at the required time-step of the simulation. On the other hand, the MEGAN model requires information of the previous day meteorological conditions. For that, several approaches are found in the literature, from implementations providing the average of the previous day temperature and solar radiation to approaches that work with monthly averages. MEGAN shows important sensitivity on these approaches as described in the manuscript: "[...]Marais et al. (2014) performed several sensitivity model runs to study the impact of different model input and settings on isoprene estimates that resulted in differences of up to 17% compared to a baseline". In our study, weather inputs are based on previous day 24h averages and data for the hour of interest.*

*Marais, E. A., Jacob, D. J., Guenther, A., Chance, K., Kurosu, T. P., Murphy, J. G., Reeves, C. E., and Pye, H. O. T.: Improved model of isoprene emissions in Africa using Ozone Monitoring Instrument (OMI) satellite observations of formaldehyde: implications for oxidants and particulate matter, Atmos. Chem. Phys., 14, 7693-7703, doi:10.5194/acp-14-7693-2014, 2014.*

P 9

L3: better say "every 720 s"

*Response: Amended.*

L 5: Which fields are initialised (also clouds or only T, v,w,q). What is known about the biases of the 24 h forecasts?

*Response: The meteorological model NMMB is initialized with the Final Analysis (FNL) of NCEP. The fields initialized are temperature, winds, specific humidity, and cloud water mixing ratio. The work developed at NCEP shows that the Anomaly Correlation at 500 hPa at 24 h of forecast is nearly 1, see Figure 4. This provides clear evidence of the skill of the model in NWP. The skills are comparable to those of GFS.*

[Figure]

*Figure 4. Anomaly Correlation of the NMMB (green lines) and GFS (black lines) 9-day forecast of the 500 hPa Height.*

L 12: not using 2004 fire emissions is a severe omission (see my general comment). Please clarify what fire emissions have been used. Was it an average for the period? It is not clear what "interpolated" means. Do the fire emissions have a seasonal cycle ?

*Response: The anthropogenic emissions and biomass burning emissions used in our work are those described in Lamarque et al. (2010), to allow an easy intercomparison. An error*

*was introduced in the initial manuscript, because the emissions are not computed for the year 2004; we use the emissions derived by Lamarque et al. (2010) for the 2000. The details on how these emissions were derived can be read in Lamarque reference. Now, in the revised manuscript the sentence "Note that the 2004 emissions are derived from a linear interpolation between years 2000 and 2010" has been changed to "Note that this methodology involves assuming 2004 emissions equivalent to the best estimate reported by ACCMIP for year 2000". The biomass burning emissions provided by Lamarque et al. (2010) have a seasonal cycle, while anthropogenic emissions are constant throughout the year.*

*Lamarque, J.F., Bond, T.C., Eyring, V., Granier, C., Heil, A., Klimont, Z., Lee, D., Liousse, C., Mieville, A., Owen, B. and Schultz, M.G., 2010. Historical (1850–2000) gridded anthropogenic and biomass burning emissions of reactive gases and aerosols: methodology and application. Atmospheric Chemistry and Physics, 10(15), pp.7017-7039.*

L 26: see my general comment, please justify the choices
**Response:** *The injection height of the different emissions used in our model are derived from sensitivity runs. Please, see previous discussion in the general comments about the justification of the emission heights.*

L 29: Please provide reference the strong impact of aerosol on the photolysis rates.
**Response:** *Real and Sartelet (2011 ACP) studied the effect of aerosols in the photolysis rates and gaseous species. They state that "Differences in photolysis rates lead to changes in gas concentrations, with the largest impact simulated on OH and NO concentrations. At the ground, monthly mean concentrations of both species are reduced over Europe by around 10 to 14% and their tropospheric burden by around 10%. The decrease in OH leads to an increase of the lifetime of several species such as VOC". ". On the other hand, Bian et al. (2003) evaluated the effect of aerosols on the global budgets of O3, OH and CH4 through their alteration of photolysis rates. The impact identified was to increase tropospheric O3 by 0.63 Dobson units and increase tropospheric CH4 by 130 ppb (via tropospheric OH decreases of 8%). Although the CH4 increases were global, the changes in tropospheric OH and O3 were mainly regional, with the largest impacts in northwest Africa for January and in India and southern Africa for July. We have included these references in the revised manuscript.*

*Real, E., and K. Sartelet. "Modeling of photolysis rates over Europe: impact on chemical gaseous species and aerosols." Atmospheric Chemistry and Physics 11.4 (2011): 1711-1727.*
*Bian, H., M. J. Prather, and T. Takemura, Tropospheric aerosol impacts on trace gas budgets through photolysis, J. Geophys. Res., 108(D8), 4242, doi:10.1029/2002JD002743, 2003.*

P 11
L27: It is not clear how missing data in the surface observations were considered. If you compare only averages without timely match give numbers of the amount of missing data.
**Response:** *If missing data are found in the observations, the corresponding time period is removed from the model data. Following this approach, model and observation derived averages are fully comparable (as only matching periods are used for the evaluation). This is clarified in the manuscript as follows: At the surface-level, daily O3 averages are computed from 3-hourly values of model and observations, applying a filter to model data whenever observations are missing, so as to consider timely/collocated values. Section 1 of the supplementary material presents the statistical measures calculated from the daily data.*

L 30: 1000m asl.? This could be a mountain stations near to the coast or a station on a flat plateau inland. It would be better to include the model orography in the choice of the mountain stations. (say 500 m above orography)

*Response: The altitude 1000 m asl. has been shown as a clear transition from a boundary-layer to a free-tropospheric regime for ozone (Chevalier et al., 2007). We are interested in filtering mountain stations because they provide measurements that are normally representative of the local conditions, but not of the air parcel that would be included on a model grid cell, of 1.4°x1°. Applying this threshold for station filtering is a common practice for several works devoted to atmospheric chemistry-transport model evaluation (Solazzo et al., 2012; Solazzo and Galmarini 2016).*

*Chevalier, A., Gheusi, F., Delmas, R., Ordóñez, C., Sarrat, C., Zbinden, R., Thouret, V., Athier, G., and Cousin, J.-M.: Influence of altitude on ozone levels and variability in the lower troposphere: a ground-based study for western Europe over the period 2001–2004, Atmos. Chem. Phys., 7, 4311-4326, doi:10.5194/acp-7-4311-2007, 2007.*

*Efisio Solazzo, Roberto Bianconi, Robert Vautard, K. Wyat Appel, Michael D. Moran, Christian Hogrefe, Bertrand Bessagnet, Jørgen Brandt, Jesper H. Christensen, Charles Chemel, Isabelle Coll, Hugo Denier van der Gon, Joana Ferreira, Renate Forkel, Xavier V. Francis, George Grell, Paola Grossi, Ayoe B. Hansen, Amela Jeričević, Lukša Kraljević, Ana Isabel Miranda, Uarporn Nopmongcol, Guido Pirovano, Marje Prank, Angelo Riccio, Karine N. Sartelet, Martijn Schaap, Jeremy D. Silver, Ranjeet S. Sokhi, Julius Vira, Johannes Werhahn, Ralf Wolke, Greg Yarwood, Junhua Zhang, S.Trivikrama Rao, Stefano Galmarini, Model evaluation and ensemble modelling of surface-level ozone in Europe and North America in the context of AQMEII, Atmospheric Environment, Volume 53, June 2012, Pages 60-74, ISSN 1352-2310, http://dx.doi.org/10.1016/j.atmosenv.2012.01.003.*
*Solazzo, E. and Galmarini, S.: Error apportionment for atmospheric chemistry-transport models – a new approach to model evaluation, Atmos. Chem. Phys., 16, 6263-6283, doi:10.5194/acp-16-6263-2016, 2016.*

P 12
L1: This choice of the tropopause is not consistent with the choice of the tropopause for the chemical boundary conditions (P7L13).

*Response: We agree with the reviewer's comment. However, one has to bear in mind that $NO_x$ concentrations in the tropopause region are very low, as the stratospheric $NO_x$ layer is located considerably higher in the stratosphere, and pollution $NO_x$ is mostly located closer to the surface. Therefore the tropospheric column does not significantly depend on the choice of tropopause level. In our work, we have considered the tropopause definition as the model level interface corresponding to approximately 100 hPa in the tropics and 250 hPa in the extratropics following Horowitz et al. (2003). In this sense, we don't think that this inconsistency has a real impact in the results discussed.*

*Horowitz, L. W., Walters, S., Mauzerall, D. L., Emmons, L. K., Rasch, P. J., Granier, C., Tie, X., Lamarque, J.-F., Schultz, M. G., Tyndall, G. S., Orlando, J. J., and Brasseur, G. P.: A global simulation of tropospheric ozone and related tracers: Description and evaluation of MOZART, version 2, Journal of Geophysical Research: Atmospheres, 108, doi:10.1029/2002JD002853, http://dx.doi.org/10.1029/ 2002JD002853, 2003.*

L 29: see my general comment. The aerosol effect may not be the most important one. There are many other possible explanations: high CH4, water vapour, clouds and photolysis, excessive mixing of emissions etc.

*Response: The authors agree with the reviewer comment. In this sense, the sentence of L29 has been reformulated in the revised manuscript in the following way: "Therefore, the*

*lack of lightning emissions in our model run could at least partly explain the lower OH values above 500 hPa reported here. Another potential explanation is the lack of aerosols in our simulation, which may overestimate photolysis rates in polluted regions (e.g., Bian et al., 2003; Real and Sartelet, 2011)."*

P 13
L1: CH4 is also a CO source. Please also reformulate the sentence.
*Response: CH4 is an hydrocarbon, so as stated in the manuscript, the photolysis of hydrocarbons are a source of CO. For clarification, "including methane" is now written in the revised manuscript.*

L 14 add reference for C-IFS
*Response: Amended.*

L 29: Could the high methane be a reason ?
*Response: The influence of $CH_4$ on modeled CO levels has been assessed through a sensitivity test, which resulted in a low impact (please, see our previous response). Changing the $CH_4$ prescribed value from 1.85 ppm to 1.78 ppm (2004 global average as reported by WMO) leads to changes in daily average CO concentration up to ±0.12 ppb, which leads us to believe that other factors have a larger impact on CO burden (see for instance Shindell et al., 2006).*

*Shindell, D. T., et al. (2006), Multimodel simulations of carbon monoxide: Comparison with observations and projected near-future changes, J. Geophys. Res., 111, D19306, doi:10.1029/2006JD007100*

L 30: Figure 3. Please show separate plots for NH, SH mid-latitudes and tropics. The seasonality is obscured by averaging over all stations.
*Response: We have only 14 stations available for CO, and only 2 of them are located in the SH. For that reason, we decided to average over all stations. The seasonality for the CO is represented in Figure 6.*

P 14
L2: no seasonality of the anthropogenic emissions is an oversimplification.
*Response: This limitation has been highlighted in the revised manuscript.*

L3: Figure 3 shows the relative bias (%), not MB as defined in the supplement.
*Response: Figure 3 shows the MB, not the relative bias (%). This information is now corrected in the revised manuscript.*

L3 Please clarify correlation of which time scale is shown, i.e. of the hourly, daily monthly values? Did you filter out seasonality? How important is the diurnal cycle to the correlation.
*Response: Time scale is described in the Model evaluation section: "For the surface-level comparison, three-hourly averages from the observations and model are used to compute daily O3 averages and calculate the statistical measures defined in section 1 in the supplemental material". We didn't filter out seasonality.*

L 28 Stein et al. and many other authors find a general underestimation in winter and spring NH.

*Response: Amended.*

P 15

L8: What do you mean by overestimated emissions above the PBL ?

*Response: As we described in the document, all the land-based anthropogenic emissions are emitted in the first 500 m of the model and biomass burning emissions from forests in the first 1300 m. When the PBL is lower than this altitude, emissions above the PBL can be overestimated and contribute to the positive bias identified in the CO.*

L 8: Please discuss the role of convection

*Response: The following sentence has been included in the revised manuscript: "Excessive vertical mixing by moist convection may explain the overestimation in the tropics".*

L 32: What regime (rural, urban) was used ?

*Response: Only rural stations were used in this study. This information is now included in the revised manuscript.*

P 16

L 8: Please discuss also PBL mixing during the night

*Response: We have included the following sentence in the revised manuscript: "Also, excessive mixing within the PBL during the night could contribute to a decrease in ozone titration by NO and partially explain the bias".*

P 17

L 8: see my general comment on the use of "good agreement"

*Response: More quantitative statements have been included in the revised manuscript supporting the qualitative statements initially used.*

L 17: Please mention the value of the biases.

*Response: The average wet deposition rates for the model and the observations are shown in Fig S4 (supporting material). Wet deposition MB for Europe, USA and Asia are - 200.70 mg N/m$^2$, - 36.87 mg N/m$^2$ and -163.27 mg N/m$^2$, respectively. These biases are now mentioned in the revised manuscript.*

L 26: What is the seasonal cycle of the biomass emissions ?

*Response: The ACCMIP inventory has monthly variations for biomass burning. This is described in Lamarque et al. (2010).*

*Lamarque, J.F., Bond, T.C., Eyring, V., Granier, C., Heil, A., Klimont, Z., Lee, D., Liousse, C., Mieville, A., Owen, B. and Schultz, M.G., 2010. Historical (1850–2000) gridded anthropogenic and biomass burning emissions of reactive gases and aerosols: methodology and application. Atmospheric Chemistry and Physics, 10(15), pp.7017-7039.*

P 18

L 1 "rural" (?) perhaps better remote

*Response: Amended.*

L 4: dominated by the tropics - perhaps simply because they are the largest region on earth (?)

*Response: This point is now included in the revised manuscript.*

L 7: TM5 has a similar chemical mechanism. Should it not be similar ?

*Response: NMMB/BSC-CTM and TM5 have a similar chemical mechanism. However, other processes that are represented differently in both models, such as the deposition, vertical mixing and emissions, also have an impact on the O3 burden. These processes can explain the differences in the O3 burdens between the models.*

L 16: Please clarify how the STE is calculated in your model.

*Response: The stratosphere-troposphere ozone exchange flux is calculated as the annual balance of the ozone mass crossing the 100 hPa height. This approach is accurate on the global and long-term average (Hsu et al., 2005). However, there are considerable differences among the models for calculating or specifying the stratosphere-troposphere exchange of ozone. This is now indicated in the revised manuscript.*

*Hsu, J., M. J. Prather, and O. Wild (2005), Diagnosing the stratosphere-to-troposphere flux of ozone in a chemistry transport model, J. Geophys. Res. ,110, D19305, doi:10.1029/2005JD006045.*

L 27: "all day long " ? Do you mean "throughout the year"

*Response: Yes. The expression has been changed to "throughout the year".*

L 27: For global models the values are commonly given in volume mixing ratios (ppb). Try to avoid mg/m3 throughout the paper.

*Response: Most of the surface observations collected are in $\mu g/m^3$. For that reason, we prefer to keep the measurement units when surface comparisons are done.*

L 28: The emission injection (500m) leads to a dilution of NO and therefore a reduction of the ozone titration. This could also explain the overestimation.

*Response: The authors agree with the reviewer's comment. In the revised manuscript this possible explanation of the overestimation of O3 is included.*

P 19
L4 Please quantify biases, what do you mean by "error".

*Response: In the revised manuscript the biases are now quantified and the sentence "error" has been substituted by "root mean square errors" to clarify the text. The quantification of the biases is provided in Figure 12.*

L 15. This points to biases of the COPCAT ozone, which has consequences for the photolysis rates.

*Response: The authors agree with the reviewer on the possible effect of the COPCAT biases upon the tropospheric ozone. This has been introduced in the revised manuscript.*

P 20
L 8: The lack of aerosol modulation of photolysis is a probably a minor aspect. Lack of heterogeneous chemistry (N2O5) might be more important. Please also mention the main

shortcomings of this simulation: (1) no lightning, (2) no 2004 biomass burning emissions and (3) no seasonal cycle for anthropogenic emissions.

*Response: The authors consider that the effect of aerosols on photolysis and radiation is not a minor aspect, as the results in Real and Sartelet (2011 ACP) indicate: "Differences in photolysis rates lead to changes in gas concentrations, with the largest impact simulated on OH and NO concentrations. At the ground, monthly mean concentrations of both species are reduced over Europe by around 10 to 14% and their tropospheric burden by around 10%. The decrease in OH leads to an increase of the lifetime of several species such as VOC". Furthermore, Bian et al. (2003) evaluated the effect of aerosols on the global budgets of O3, OH and CH4 through their alteration of photolysis rates. The impact identified was to increase tropospheric O3 by 0.63 Dobson units and increase tropospheric CH4 by 130 ppb (via tropospheric OH decreases of 8%). Although the CH4 increases were global, the changes in tropospheric OH and O3 were mainly regional, with the largest impacts in northwest Africa for January and in India and southern Africa for July. We have included these references in the revised manuscript.*

*Following the reviewer's suggestion now the main shortcomings of the simulation presented in the manuscript are clearly stated. The following sentence has been added in the conclusions section: "We note that in this contribution, we omitted aerosols and lightning emissions; anthropogenic emissions disregard seasonality; and biomass burning emissions are not specific to 2004."*

*Real, E., and K. Sartelet. "Modeling of photolysis rates over Europe: impact on chemical gaseous species and aerosols." Atmospheric Chemistry and Physics 11.4 (2011): 1711-1727.*
*Bian, H., M. J. Prather, and T. Takemura, Tropospheric aerosol impacts on trace gas budgets through photolysis, J. Geophys. Res., 108(D8), 4242, doi:10.1029/2002JD002743, 2003.*

L15: The paper provides no evidence for this claim - it can therefore not be a conclusion.

*Response: The authors do not present as a conclusion the effect of aerosols on the concentrations of OH. We only suggest some possible reasons to explain why our results of OH at northern latitudes are slightly higher than the climatological mean of Spivakovsky et al. (2000). We agree with the reviewer that this cannot be a conclusion of the work, in this sense, we only highlight a possible explanation that deserves further research in the future studies with the model.*

L 34 see above, no evidence in the paper

*Response: Addressing the reviewer's comment, the following statement has been removed from the revised manuscript: "CO production from VOCs biogenic emissions, calculated online and depending on meteorological variables such as radiation, might be overestimated too, due to the lack of aerosol attenuation of radiation".*

P 21
L 13 better "megacities"
*Response: Amended.*

Figure 3: better to have plots for different regions (NH, SH, Tropics) to better see the seasonal cycle. Choose a smaller y-range for more clarity. Show plot in ppb (as for the profiles) rather than microgramm/m3.

*Response: We have only 14 stations available for CO, and only 2 of them are located in the SH. For that reason, we decided to average over all stations. The seasonality for the CO is represented in Figure 6. Most of the surface observations are in $\mu g/m^3$ and therefore we used $\mu g/m^3$ in the surface comparison.*

Figure 4: MB (see supplement) is defined without scaling (i.e. not relative in %). Please show the MB as defined in the supplement. Use ppb as unit.
*Response: In the document the MB is calculated, not the relative bias (%). This information is now corrected in the revised manuscript. Surface observations are shown in $\mu g/m^3$.*

Figure 7: better y-range, use ppb
*Response: Surface observations are shown in $\mu g/m^3$.*

Figure 8: as for Figure 4
*Response: In the document the MB is calculated, not the relative bias (%). This information is now corrected in the revised manuscript.*

Figure 11: use ppb
*Response: Surface observations are shown in $\mu g/m^3$.*

Figure 12: see Figure 4
*Response: In the document the MB is calculated, not the relative bias (%). This information is now corrected in the revised manuscript.*

Figure 13: choose x-range 0-100 for better clarity in the troposphere.
*Response: The x-range 0-100 has been chosen to include both the tropopause and the lower stratosphere.*

Figure 14: see Figure 4
*Response: In the document, the MB is calculated, not the relative bias (%). This information is now corrected in the revised manuscript.*

---

## Referee Report (RR1)

Review of

"Gas-phase chemistry in the online multiscale NMMB/BSC Chemical Transport Model: Description and evaluation at global scale" by Badia et al.

This is the review of the revised manuscript.

Remarks

The author have provided a comprehensive response to my first review. Although I still think lightning emissions should not be neglected in a global model run, I am OK to accept the authors' justification for this if it is made in clear in the paper. I recommend stating this fact in the abstract.

Please indicate also in the response what you are going to change in the text. Without this information it is difficult for the reviewer to judge how the actual manuscript has been amended.

No tracked changes were indicated in the reference section but I believe that at least some new references appeared. So please mark them in new response.

I found that too few aspects of the response were actually included in the revised paper (or it was perhaps not marked as a track change). I would welcome if some of the discussion would appear also in the paper. So please briefly summarize the references on the impact of lightning NO, aerosol impact on photolysis rates, the performance of the COPYCAT scheme, the meteorological performance of the model and the emission injection profiles.

I asked for a better quantification for statements such as "good agreement". I found a few examples where this has been done but there are still hardly any quantification in the conclusion. Please include a quantification next to the qualitative statements in the conclusion.

I suggested to use volume mixing ratios rather concentrations (microgram/m3) also for the evaluation with surface data. This was done not only for the reasons of consistency. Concentrations depend on density and therefor on the station height. Differences in the model orography and station height have therefore an impact on the evaluation results (beside the scientific problem of the vertical representativeness). This could be avoided by converting the concentrations in volume mixing ratios using the stations measurements of p and T. I recommend using ppb as unit for the comparison.

The wording and clarity of the paper has been improved a lot.